# Natural hazard events affecting transportation networks in Switzerland from 2012 to 2016

Jérémie, Voumard[1], Marc-Henri, Derron[1], Michel, Jaboyedoff[1]

[1] Risk analysis group, Institute of Earth Sciences, FGSE, University of Lausanne, Switzerland

## Abstract

Switzerland is threatened by many natural hazards. Many events occur in built environments, affecting infrastructure, buildings and transportation networks, occasionally producing expensive damages. This expense is why large landslides are generally well-studied and monitored in Switzerland to reduce the financial and human risks. However, there is a lack of data on small events, which have recently affected roads and railways. Therefore, in this study, all of the reported natural hazard events that affected Swiss transportation networks since 2012 were collected in a database. More than 800 events affecting roads and railways were recorded within in a five-year period from 2012 to 2016. These events are classified into six classes: earth flow, debris flow, rockfall, flood, snow avalanche and "others."

Data from Swiss online press articles were sorted by Google Alerts. The search was based on more than thirty keywords in three languages (Italian, French and German). After verification that the article was related to an actual event that affected a road or a railway track, it was studied in detail. We collected information on more than 170 attributes of events, such as the event date, event type, event localization, meteorological conditions, impacts and damages on the track and human damages. From this database, a variety of trends over the five-year period can be observed in the event attributes, particularly the spatial and temporal distributions of the events, and their consequences on traffic (closure duration, deviation, costs of direct damage).

The database is imperfect due to the short period of data collection, but it highlights the non-negligible impact of small natural hazard events on roads and railways in Switzerland at a national level. This database contributes to understanding and quantification of these types of events and better integration in risk assessment.

## Keywords

natural hazard events, floods, landslides, earth flows, rockfalls, debris flows, snow avalanches, transportation networks, Switzerland, database

## 1 Introduction

Natural hazards cause many damages to transportation networks worldwide (Nicholson & Du, 1997; Hungr et al., 1999; Dalziell & Nicholson, 2001; Karlaftis et al., 2007; Tatano et al., 2008; Erath et al. 2009; Muzira et al., 2010; Jelenius et al., 2012). Particularly in mountainous areas, floods, landslides (considered earth flows in this study), debris flows, rockfalls and snow avalanches (called avalanches in this paper) can seriously affect the traffic on roads and railway tracks, isolating villages or regions and generating infrastructure and economic damages (Bunce et al., 1997; Budetta et al., 2004; Evans et al., 2005; Collins, 2008; Salcedo et al., 2009; Guemache et al., 2011; Jaiswal et al., 2011; Michoud et al., 2012; Laimer, 2017b).

Large natural hazard events affecting roads and railways are generally well studied and documented, e.g., the Séchilienne landslide (Kasperski et al, 2010), La Saxe landslide (Crosta et al. 2014) or La Frasse landslide (Noverraz and Parriaux, 1990), but this is not the case for minor and medium-sized events with deposit material on the track ranging from a few cubic decimetres to a few thousand cubic metres. They are numerous and often too small, making them difficult to detect and expensive to monitor (Jaboyedoff et al. 2016a).

Generally, disasters events or events with any high social impact (death, high cost, highlighting societal problems, etc.) are collected in a database. The criterion to be listed in the main global disaster databases (EMD-DAT, Swiss Re, Dartmouth) illustrate this because at least ten casualties or other political or economic criteria are required (Guha-Sapir et al., 2015; Swiss Re, various dates; Dartmouth Flood Observatory, 2007). Insurance databases are more detailed, however, they are usually not publicly available, such as the NatCat from Munich Re reinsurance (Tchögl et al, 2006; Bellow et al., 2009; Munich R. E., 2011). At present, most worldwide, national and regional databases do not generally include small events that are considered insignificant to experts (Guzzetti et al. 1994, Malamud et al. 2004; Petley et al. 2005; Devoli et al. 2007; Kirschbaum 2010, Foster et al. 2012; Damm et al. 2014). There are also noteworthy exceptions such as the RUPOK database (Bíl et al. 2017), which collects information about the consequences of geohazards on transportation networks. The Swiss flood and landslide damage database (Hilker, 2009) contains small events,

although events with direct damage costs less than EUR 8 500 are not considered. Moreover,
there is no information about track and traffic effects.
Gall et al. (2009) highlighted that underreporting of small events induces bias in data. The
director of the Global Resource Information Database at the UNEP recognized a problem in
evaluating the true impact of natural hazards because the EMD-DAT database only records
events with estimated losses greater than 100 000 US$ (Peduzzi, 2009). The Head of the
UNISDR, R. Glasser, notes that governments underestimate low-cost disasters that
significantly affect societies (Rowling, 2016).
To fill a gap in the knowledge about small events, in this study, we focused on the impacts of
natural hazards on roads and railway tracks, collecting as much information as possible on the
events that affected the Swiss transportation network since 2012.
The goal of this database is to determine the main trends of these events and evaluate the
relevance of concerns.

## 74  2   Study area

The study is applied to all of Switzerland, which has a surface area of 41 285 km$^2$, with an
elevation ranging from 193 m (Lake Maggiore) to 4 634 m a.s.l. (Dufourspitze). The Swiss
geography can be divided into three major geomorphologic-climatic regions: the Alps, the
Plateau and the Jura. The Alps cover 57% of the Swiss territory (23'540 km$^2$) with 48
summits over 4 000 m a.s.l. and many inhabited valleys. The Plateau, located northwest of the
Alps, covers 32% of the territory (13 360 km$^2$) at an average altitude of approximately 500 m
a.s.l. and is partially flat with numerous hills. Two-thirds of the Swiss population lives on the
Plateau (13 360 km$^2$), which has a population density of 450 inhabitants per square kilometre.
The Jura Mountains (11% of the territory, 4 385 km$^2$) is a hilly and mountainous range
situated on the north-western border of the Plateau, with a top summit of 1 679 m a.s.l. (Mont-
Tendre). The Swiss climate is a mix of oceanic, continental and Mediterranean climates, and
varies greatly because of the relief. The average annual rainfall is approximately 900-1 200
mm years$^{-1}$ on the Plateau, 1 200-2 000 mm years$^{-1}$ on the Jura Mountains and 500 to 3 000
mm years$^{-1}$ in the Alps (Bär, 1971). The Swiss average temperature is approximately 5.7°C
(MeteoSwiss, 2018).

## 3   Data and methods


A database was constructed for the 5-year period of 2012 to 2016 and 846 events were
collected. The minimum threshold for inclusion in the database was a traffic disruption (for
example, a large-velocity reduction) for at least 10 minutes following a natural hazard event
that reached a transportation track.
We used online press channels as information sources because of the ratio of simplicity and
efficiency. An online press review was made every working day from 2012 to 2014; in May
2014, Google[tm] Alerts (Google, 2018) was introduced with more than fifty keywords in
German, French and Italian (see Table 1-SM in Supplementary material (SM)). These alerts
(approximately ten per day) allowed for the collection of events from the Swiss online press.
Each alert contained an average of two online press articles with one of the fifty keywords.
Each article was verified to identify whether the related information concerned a natural
hazard event that affected a transportation network. If not, it was disregarded.
Approximately 10% of all these highlighted articles referred to a real natural hazard event.
Approximately 800 articles were collected from mid-2014 until the end of 2016. The Swiss
traffic information website was also periodically manually checked, as well as several social
media pages that contained pictures of events, such as the official Facebook page of the
commune of Montreux (Montreux, 2014). In addition, some events were collected directly in
the field.
We classified natural hazards according to six categories:
- Static or dynamic flood with little sedimentation materials on the track, including a
few hail events.
- Debris flow that is often not well described in the media and confounded with
landslide or flood. It is often characterized using pictures from the press articles.
- Landslide: superficial or deep sliding of soil mass including shallow landslides.
- Rockfall refers to rock falls and rockslide.
- Avalanche refers to snow avalanches.
- "Other": snowdrifts (mainly during February 2015 in west Switzerland) and falling
trees (mainly during windstorms).
attributes were used to describe the events (Table 1; Figures 1-SM and 2-SM in the
Supplementary material (SM)) and were subdivided into eight categories: date, location, event
characterization, track characterization, damage, weather, geology and sources. Data about the
date, location, event characterization and damage were obtained from online press articles.
Attributes of the database are presented in Table 1.
Images from the press articles were used to estimate many attributes such as the event
classification and volume estimation of the deposit material, if it was not estimated or noted in
the press article.
The analyses were performed in a Geographic Information System (GIS) environment, for
spatial data, or using standard statistical methods for non-spatial data. To extract the general
trends of the 846 events collected from 2012 to 2016, the data were characterized by basic
statistics descriptors and displayed in histograms and charts.
Weather data were obtained from 24 weather MeteoSwiss stations. For each event, the
reported weather conditions were not always from the closest station; data was obtained from
a station with a similar topo-climatic situation. The average distance between weather stations
and events was 20 km (SD of 18 km) and the average absolute elevation difference was 200 m
(SD of 366 m). The rainfall data were given for the event day, the previous five days and the
last ten days to provide the antecedent situations.
The deviation lengths for roads were measured using ArcGIS. Density maps were prepared
using the kernel density function in ArcGIS with a search radius of 10 km for the events map
and 20 km for the road density map, with a 500 m output cell size for both. The results were
classified into 10 classes using the Jenks natural breaks method in ArcGIS.
The damage levels were characterized by four levels, partially based on Bíl et al. (2014). The
first damage level was "no closure or no track damage". Events of this level generate only
traffic slowdowns and small disruptions. They mainly comprise floods, often triggered by
strong storms (vehicles can drive slowly on a flooded road without the need to close the track)
(Figure 6E). The reduction of the traffic velocity generally lasts less than two hours. The
second level refers to a complete or partial track closure because of material deposition on the
track. If only one lane is closed, the second lane allows for alternated traffic moderated with
temporary traffic lights or traffic regulators. Tracks with the second level of damage can
reopen after evacuation work, without any repair work.
In addition to track closure, the third level, "partial damage", requires superficial repairs
and/or minor stabilization of the track embankments because the events resulted in small
damage to the tracks. Finally, the "total destruction" level indicates that, in addition to track
closure, the track embankment must be reconstructed, requiring significant repair work.
The costs per square metre were attributed for each damage class according to the event
intensity (small, middle and large) for both roads and railways. A surface area of deposit
material on the track of 100 m$^2$ is assumed to be a small event, 200 m$^2$ is a medium event and
300 m$^2$ is a large event. The costs are given in Euros, with the mid-January 2018 value of 1
EUR = 1.17 CHF = 1.23 USD. On average, EUR 6 per square metre was estimated for the "no
closure" class, EUR 230 for "closure", EUR 400 for "partial damage", EUR 1 000 for "total
destruction" and EUR 230 for the "unknown" class (Table 2-SM). Direct damage cost
evaluation was based on road and railway reports (Canton de Vaud et du Valais, 2012; SBB
CFF FFS, 2017) and on repair work cost provided by an experienced Swiss civil engineer.
Direct damage costs are difficult to assess (even more so for indirect damage costs), thus the
proposed methodology to determine them must be considered a tool to compare the costs of
the different damage classes. The cost values should not be considered as the true costs for all
events but as an order of magnitude of the costs (see section 5.4).
*Table 1: Attribute categories describing events in the database.*

| Attribute category | Question | Contains | Number of attributes | Main source |
|---|---|---|---|---|
| ID | Event ID | - | 1 | - |
| Date | Which date and time | Year, season, day part | 15 | Online press article |
| Location | Where did the event occur? | Region, topography, coordinates | 21 | Online press article and GIS[1] |
| Event characterization | Which natural hazard event? | Type of hazard, features, picture | 12 | Online press article |
| Track characterization | On which track? | Road/railway, features, deviation | 17 | Swisstopo[2] |
| Damage | Which kind of damage? | Damage on track, vehicle, people | 11 | Online press article |
| Weather | What was the weather? | Sun, rain, temp., storm, wind, snow | 68 | MeteoSwiss[3] |
| Geology | On what soil did it occur? | Soil features | 11 | Swisstopo[2] |
| Source | What are the information sources? | Addresses of online press articles | 16 | Online press article |

[1] GIS: Geographic Information System
[2] Swisstopo: Swiss Federal Office of Topography
[3] MeteoSwiss: Swiss Federal Office of Meteorology and Climatology

# 4 Results

## 4.1 Types of natural hazard processes

50% (421 events) of the 846 collected events are floods, including 1% (8 events) hail flooding events (Figure 1A). The second most frequent process was landslides (23%; 192 events), followed by rockfalls (11%; 96) and debris flows (8%; 68). The remaining were avalanches (2%; 15), and "other" processes (6%; 54) including snowdrifts (4.5%; 40) and falling trees (1.5%; 14). Snowdrifts mainly resulted from a unique event in February 2015.

## 4.2 Spatiotemporal conditions

### 4.2.1 Spatial distribution

Natural hazard events affecting the Swiss transportation network from 2012-2016 were equitably distributed over the geomorphologic-climatic regions of the Plateau and Alps (44% each; 371 and 377 events, respectively). The remaining 12% (98 events) occurred in the Jura area (Figure 1B and Figure 2 and; Table 3-SM). The spatial distribution of natural hazard events beside floods was proportional to the surface areas of Swiss regions: the Alps, with 60% of the Swiss territory surface, account for 64% of events except floods, the Plateau for 30% and 31%, and the Jura for 10% and 5% respectively. The kernel density maps of all event types and the road density map are shown in Figure 2-SM.

The majority of the floods (57%; 239 events) occurred in the Plateau. Debris flows occurred mostly in the Alps (96%; 66), as well as rockfalls (88%; 84) and avalanches (100%; 16), which is not surprising considering the presence of steep slopes. Landslides are more equally distributed, with only 55% (107) in the Alps because they usually occur on moderate slopes (Stark and Guzzetti, 2009). The "other" events (snowdrift and falling trees) occurred mostly on the Plateau (41; 79%).

Almost half of the events (49%; 412 events) occurred in a built environment (towns, agglomerations, villages and hamlets) and approximately half (51%; 434) of events occurred in a natural environment (countryside: 25%, 211; forest: 22%, 185; and mountain above the forest limit: 4%, 38) (Figure 1C; Table 4-SM).

In the risk ratios (Miettinen, 1972; Zhang and Kai, 1998; Spiegelman and Hertzmark, 2005) related to the surface of the regions, floods and "other" are over-represented in the Jura and in Plateau whereas debris flows, avalanches and rockfalls are over-represented in the Alps

(Figure 3A). The risk ratio related to the length of the roads of the three regions indicates that
the Alps have over-represented debris-flows, landslides, rockfalls and avalanches (Figure 3B)
The slope angle distribution (Figure 1D; Table 5-SM), extracted from a 25 m DEM
(Swisstopo, 2018), indicates that 40% (339 events) of all events affected tracks on slopes
from 0° to 5° and that 30% (257 events) occurred from 5° to 15°. 62% (260 events) of floods
affected tracks on an almost flat slope, from 0° to 5°, and 43% (30 events) of debris flows
occurred on a 5°-15° slope. A third of landslides (63 events) and a third of rockfalls (30
events) occurred on a 15°-25° slope. 76% (12 events) of avalanches crossed tracks on a slope
angle of 10°-30°. Two-thirds (36 events) of "other" processes were observed on a 0° to 5°
slope.
Based on the Swisstopo maps, eight slope orientations were estimated to account for 72%
(609 events) of the recorded events (Figure 3-SM). Slopes oriented to south, south-east and
west accounted for 17% (144 events) each. The over-representation of these orientations is
caused by debris flows occurring on the western slopes (mainly due to debris flows that
occurred in the S-Charl valley in 2015). Landslides appeared to occur more often on south-
and west-oriented slopes.

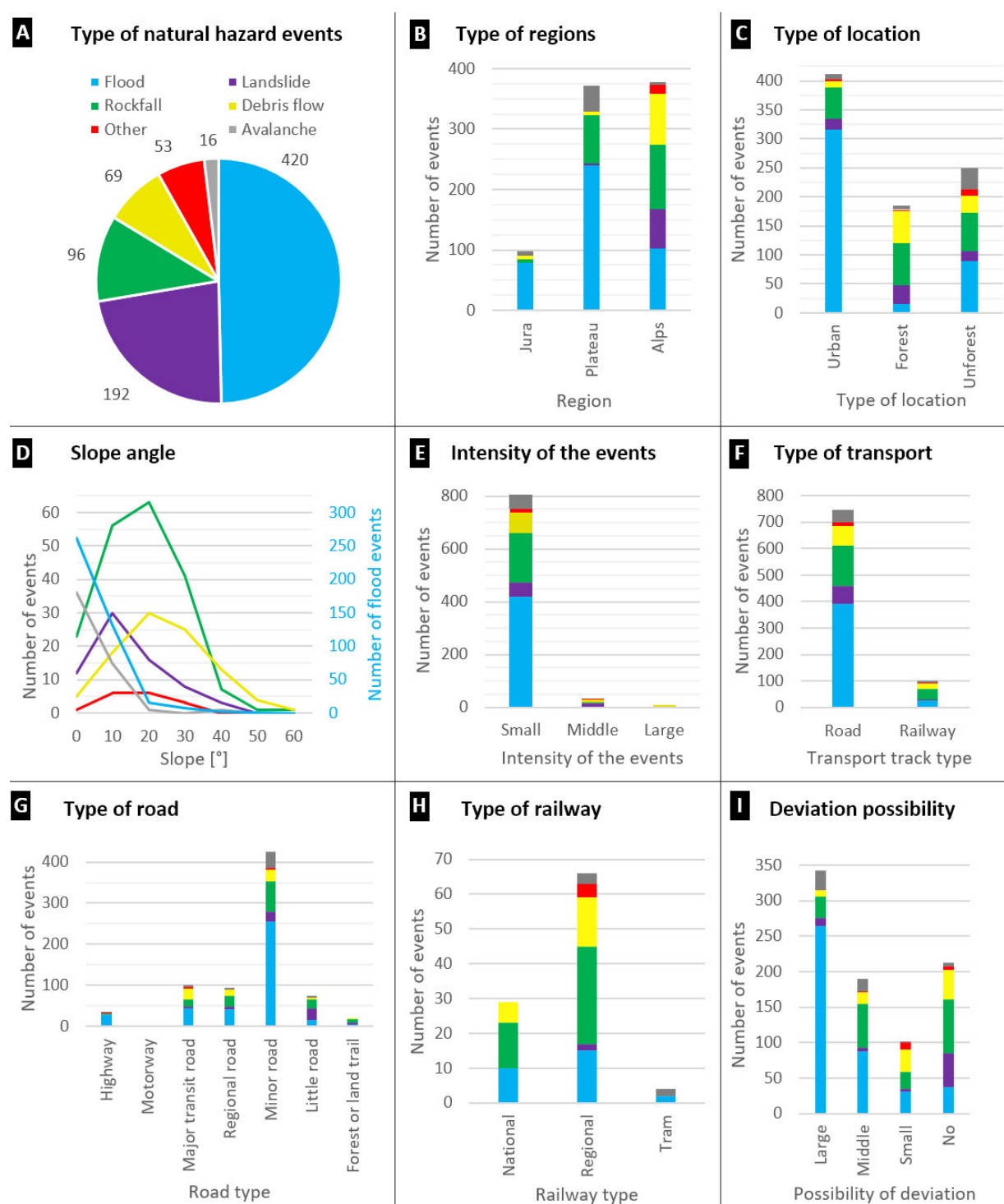

Figure 1: A: Number natural hazard events on the Swiss transportation network from 2012 to 2016. B: Distribution throughout the three large geomorphologic-climatic regions. C Distribution of the type of location. D: Slope angle distribution. Flood events are on the secondary vertical axis. E: Distribution of events according to intensity of the deposit material on the track. Small event: 0-10 m³; middle event: 10-2000 m³, large event: >2000 m³. F: Transport mode distribution. G: Road type distribution. H: Railway type distribution. I: Distribution of the possibility of deviation. Large possibility of deviations: >3 possibilities; middle: 2-3, small: one possibility; no: no possibility.

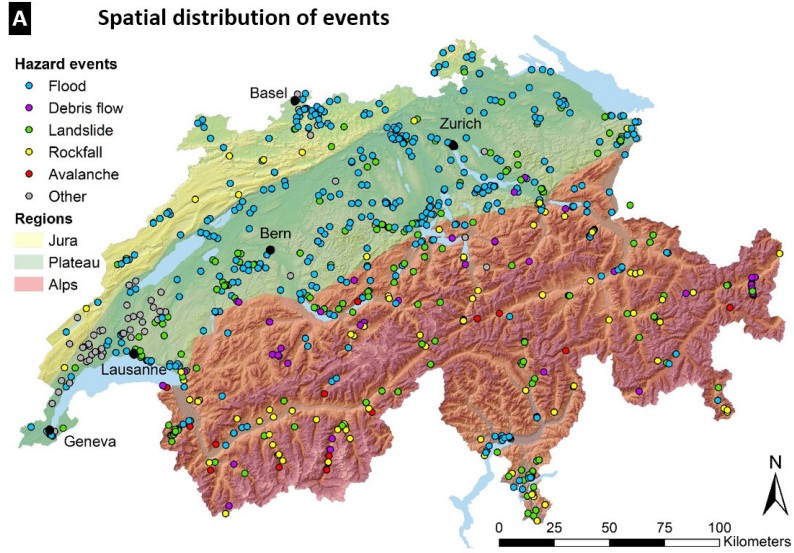

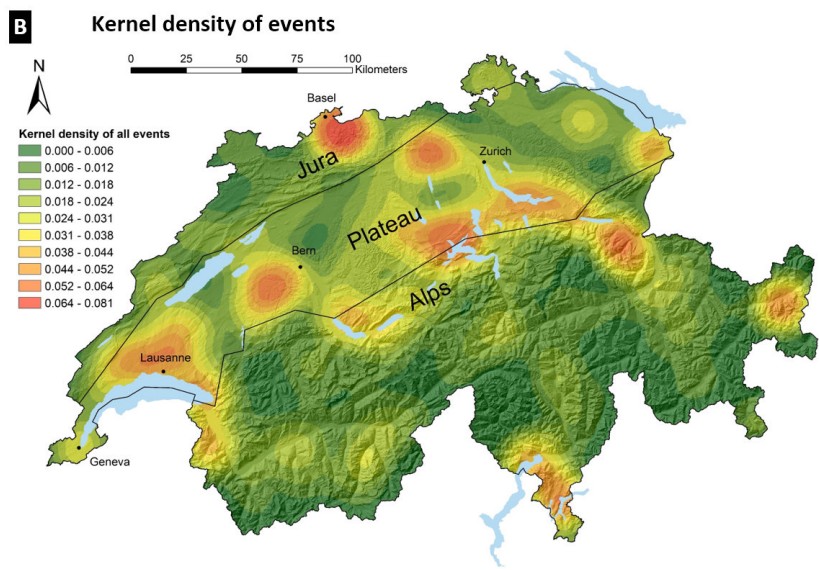


*Figure 2: A: Spatial distribution of natural hazard events affecting roads and railways in Switzerland from 2012*
*to 2016. Map source: Swisstopo. B: Kernel density of the events (20 km search radius and results classified*
*using 10 classes with the Jenks natural breaks method) based on ArcGIS functions.*

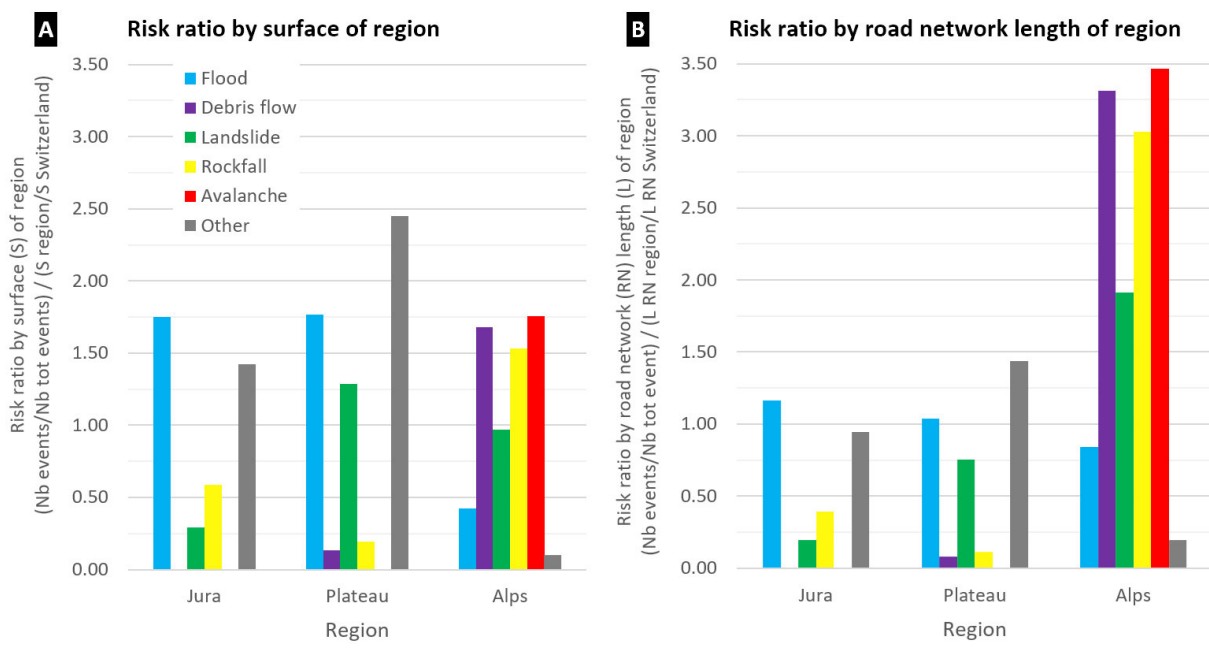


Figure 3: A: Risk ratio by surface of the three geomorphologic-climatic Swiss regions. B: Risk ratio by the road
network (RN) length of the three geomorphologic-climatic Swiss regions.



4.2.2 Event intensity
The debris flow, landslide, rockfall and avalanche events were classified into three intensity
classes (Figure 1E and Figure 4; Table 6-SM) defined by the volumes of deposit materials on
the track:
- Small: less than ten $m^3$.
- Medium: from ten to two thousand $m^3$.
- Large: larger than two thousand $m^3$.
With one exception (medium intensity), floods were classified based on the water level and
flooded area as small-intensity events (419 floods). "Other" events (snowdrifts and falling
trees) were also all categorized as small events (53 events). 95% (804 events) of the events
were classified as small, 4% (33) were medium and 1% (9) were large events. Note that a
third (32) of rockfalls were large events.
Excluding floods, 39% (146 events) of the event sources were located more than 50 m from
the track, 35% (185) were located 0 to 50 m away (Table 7-SM). A quarter (95) of the source
locations are unknown. Almost all sources close to the tracks, representing 35% (185) of all
events, can be considered human-induced natural hazard events. The sources of debris flows
and avalanches in the Alps are located far from the track and were of natural origin (100%
(69) for debris flow and 94% (15) for avalanche). Excluding floods, 80% (339) of the sources
were located above the track, 7% (29) were below the track and 14% (58) were of unknown
origin (Table 8-SM).

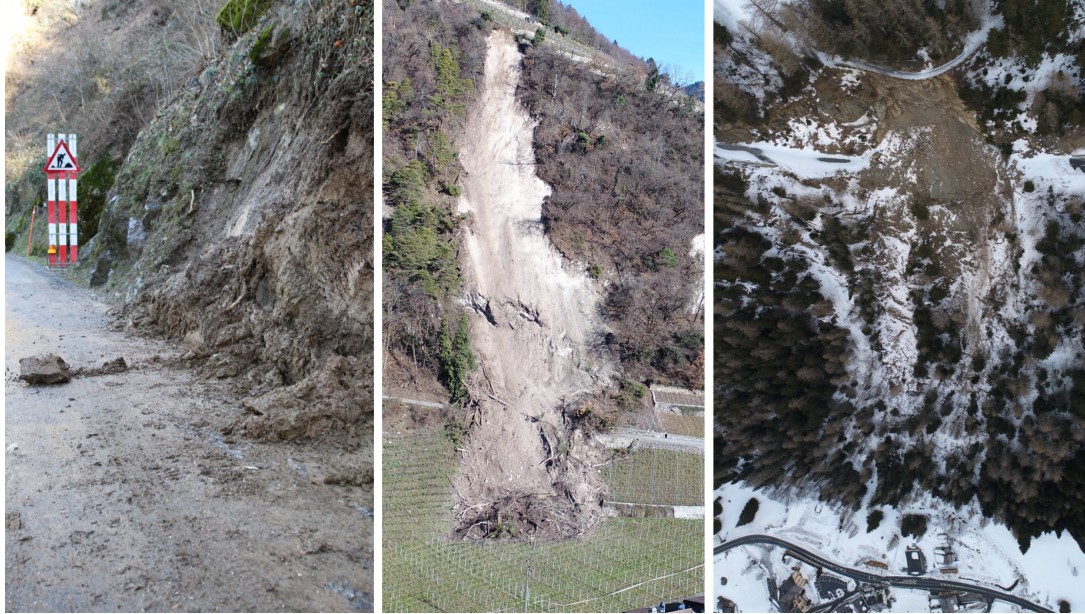

*Figure 4: Examples of events affecting roads. Left: small event on the only road to the small village of Morcles*
*(Canton of Vaud). Middle: middle-sized event on a minor road in Ollon (Canton of Vaud). Right: large event*
*with an estimated volume of 3500 m³ that cut a 50 m length on the international road between France and*
*Canton of Valais near the Forclaz pass (Trient). The road closure was estimated at six weeks. Images taken on*
*24 January 2018 after a winter storm.*

### 4.2.3 Rainfall
The average rainfall during the day of an event was 17 mm (Figure 5A; Table 9-SM). On
average, the amount of rain during the event day was 22 mm, 17 mm, 14 mm, 5 mm and 4
mm for flood, landslide, debris flow, rockfall and avalanches, respectively. The maximum
precipitation recorded (154 mm) in the database occurred in the Canton of Ticino in
November 2014, which triggered a landslide.
The debris flows mostly occurred following strong convective summer storms after a quite
sunny day. This means that the precipitation at the location of the debris flows may be higher
than those recorded by the station. Landslides occurred after the highest amount of rainfall
recorded in the last ten days preceding the event. The debris flows occurred several minutes to
a few hours after heavy precipitations, floods occurred after approximately one day of heavy
rainfall and landslides occurred up to several days after intense precipitations.

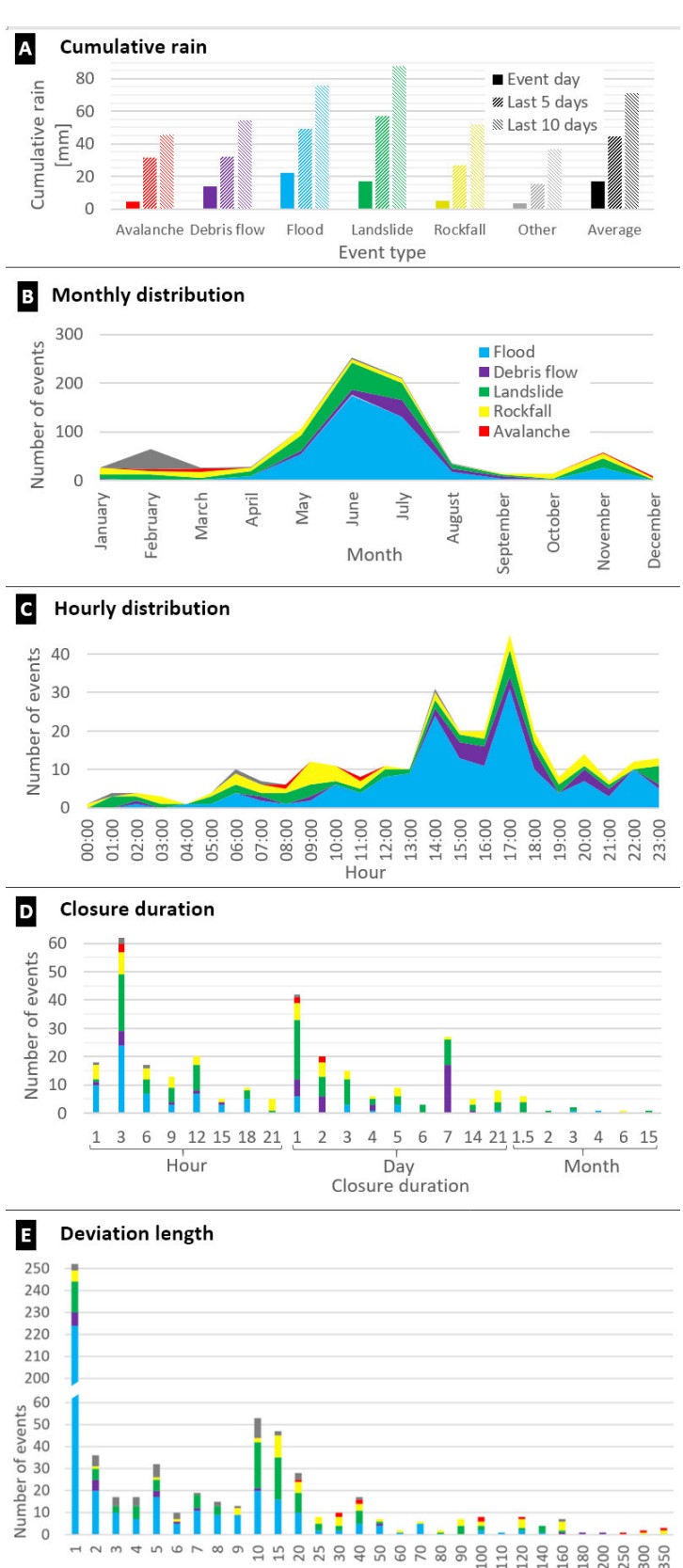


Figure 5: A: Cumulative rain [mm] distribution on the day of natural hazard events and previous five and ten
days. B: Monthly distribution. C: Hourly distribution. D: Closure duration distribution. E: Shorter deviation
length distribution of road closures. The vertical axis shows values from 60 to 200.

## 4.3  Temporal parameters
### 4.3.1  Clustering in time
Fourteen long-lasting rainfalls for a total of 111 days were selected during the five-year
period (Table 2), with durations ranging from two to fourteen days. 60% (511) of events
occurred during those 111 days of long-lasting rainfalls. Those 111 days correspond to 6% of
the total number of days over the five-year period. This highlights the negative impact of
long-lasting rainfalls, which generated an average of 4.6 events per day. A third of these 511
events were among the 50 major loss events worldwide, according to Munich Re Topic Geo
annual reports.
*Table 2: Long-lasting rainfalls resulting in 61% of the collected natural hazard events on the Swiss*
*transportation network from 2012 to 2016.*

| Date | Number of days | Number of events | Avg. number of events per day[2] | Munich Re event[3] |
|---|---|---|---|---|
| 2012.01.06-07 | 2 | 2 | 1 | 2012.01 |
| 2012.11.04-14 | 11 | 12 | 1.1 | - |
| 2013.06.01-03 | 3 | 26 | 8.7 | 2013.06 |
| 2014.02.15-18 | 4 | 4 | 1.0 | 2014.02 |
| 2014.06.03-12 | 10 | 10 | 1.0 | 2014.06 |
| 2014.07.04-15 | 12 | 44 | 3.7 | - |
| 2014.07.22-31 | 10 | 51 | 5.1 | - |
| 2014.11.13-18 | 6 | 35 | 5.8 | - |
| 2015.04.27-05.07 | 11 | 55 | 5.0 | - |
| 2015.06.05-15 | 11 | 75 | 6.8 | - |
| 2015.07.22-25 | 4 | 37 | 9.3 | - |
| 2016.06.02-09 | 10 | 80 | 8.0 | 2016.06 |
| 2016.06.15-25 | 14 | 49 | 3.5 | - |
| 2016.07.22-28 | 7 | 35 | 5.0 | - |
| Total | 111 | 511[1] | 4.6 | - |

[1] 60% of all events.
[2] Event number/number of days.
[3] Sources: Munich Re, 2013, 2014, 2015 and 2017.

### 4.3.2  Monthly distribution
The monthly distribution of events indicates an average of 71 events per month, with a
median value of 32. It ranged from 9 events in December to 253 events in July (Figure 5B;
Table 10-SM). Two-thirds of all events (68%; 570 events) occurred during the three months
of May (13%; 107), June (30%; 253) and July (25%; 210).
85% (357 events) of floods and 64% (123) of landslides occurred from May to July. 89% (61)
of debris flows occurred from May to August. 64% (61) of rockfalls occurred during the
months of January, March, May, October and November. 50% (8) of avalanches occurred in
March. 81% (43) of "other" events occurred in February.
### 4.3.3 Time of day and hourly distribution
The hour of occurrence were included for 33% (281) of the events (Figure 5C). 57% (89) of
floods with a known hour of occurrence occurred between 2 pm to 7 pm, 61% (17) of debris
flows occurred between 3 pm and 7 pm. Landslides and rockfalls were fairly well distributed
during a day; 23% (10) of rockfalls occurred between 9 and 11 am.
## 4.4 Infrastructure parameters
### 4.4.1 Types of tracks
88% (747 events) of events affected road tracks and 12% (99) have affected railway tracks
(Figure 1F; Table 11-SM). Among the events affecting roads, 53% (393) were floods, 20%
(151) were landslides, 10% (76) were rockfalls, 9% (67) were debris flows and 8% (48) were
"other" events. For the railway tracks, 42% (41) were landslides, followed by 27% (27)
floods, 20% (20) rockfalls, 5% (5) "other", 4% (4) avalanches and 2% (2) debris flows. 79%
(668) of all events occurred on minor roads or minor railway tracks and 21% (178) occurred
on major roads or major railway tracks.
The risk ratio of the number of events by transportation network type (roads or railways,
related to their respective lengths) indicates that events on railway tracks are over-represented
(risk ratio of 1.67) and under-represented on roads (0.95 risk ratio).
### 4.4.2 Roads
The Swiss road network length is approximately 72 000 km, with 1 850 km managed by the
Swiss Confederation, among which 1 450 km are highways and motorways, 25 000 km are
major (cantonal) roads and regional roads, and approximately 45 000 km of roads are
managed at the municipal level (Federal Statistical Office, 2018).
Swiss roads are classified into seven classes, according to the Swiss Federal Office of
Topography (Figure 1G: Table 12-SM). Highways have separated traffic and a speed limit of
120 km/h and motorways have a 100 km/h speed limit. Both account for 3% of the road
network length, accounting for 5% (36 events) of all events that affected roads. Major transit
roads with a high traffic load (12% of the road network length) were affected by 13% (99) of
the events. Roads of regional importance (22% of the road network length) accounted for 12%
(94) of the events with a lower traffic load, both have a maximum speed of 80 km/h. The
three remaining road classes (63% of the road network length) are based on the width of the
road and are related to small roads with low traffic. 69% (518) of events that affected the road
network were on this type of road.
Proportionate to the length of the different road types, the event frequency corresponds to one
event per 200 km per year for highways and motorways and one event per 440 km, 860 km
and 440 km per year for major, regional and minor roads, respectively. On average, roads
were affected by one event per 480 km per year.
4.4.3   Railways
The Swiss railway network is 5 400 km long, including 130 km of cogwheel train track and
202 km of tram track (Federal Statistical Office, 2018).
Railway tracks are classified into three classes: major (34% of the railway network; 1850 km),
minor (62%; 3350 km) and tram lines (4%) (CFF, 2018; Federal Statistical Office, 2018)
(Figure 1H; Table 13-SM). The major tracks usually have two lanes, linking the main Swiss
cities or crossing the Alps, and accounted for 29% (29 events) of railway events. The minor
tracks, often with one lane, were affected by two-thirds (67%; 66) of railway events. Tram
tracks in or around towns were affected by 4% (4) of railway events.
Proportionate to the length of the different track types, the event frequency along major
railways tracks was one event per 320 km per year and the minor railway tracks and tram
tracks were affected by one event per 250 km per year. On average, railway tracks were
affected by one event per 275 km per year.
4.4.4   Possibility of deviation
For each event, we determined how easy it was to find a deviation track (an alternate route to
reach the next village that avoids the closure area) (Figure 1I; Table 14-SM). For 40% (342
events) of the events, there were more than 3 possibilities of deviation. For 23% (190), there
were 1 to 3 deviation possibilities, and for 12% (102), there was only one possibility. For 25%
(212) of events, it was not possible to take an alternative track to bypass the closure because
they occurred in valleys with only one track.
91% (383 events) of flood events and 90% (48) of "other" events could be bypassed. There
were no deviation possibilities for 70% (48) of debris flows, 43% (41) of rockfalls and 40%
(77) of landslides. This indicates that it is often impossible to find a deviation path for
numerous debris flows, landslides, rockfalls and avalanches.

## 4.5 Impacts and damages

### 4.5.1 To track

80% (679 events) of all events generated track damages (Figure 6A and Table 15-SM). 18% (149) generated no closure or no track damage. 142 of those events were floods. 57% (483) of events generated track closures because of material on the tracks. In addition to closure, 17% of events (143) produced "partial damage" on the track (third damage level). The "total destruction" level accounted for 6% of all events (53). For 2% of events (18), direct damages could not be estimated.

35% (142 events) of floods caused no track closure and 62% (251) of floods generated only track closure. Floods generated the least damages. Many floods did not require track closure because vehicles or trains could pass through the water level. 39% (27) of debris flows generated partial damages and 25% (18) caused total destruction. Half (96) of landslides generated no track damages with a track closure and 39% (72) of landslides resulted in partial damage to the tracks. Half (48) of rockfalls generated only track closures and 39% (37) generated partial damages. 81% (13) of avalanches and 96% (51) of "other" events generated track closures due to the high percentage of snowdrifts (74% (39) of "other" events were snowdrifts).

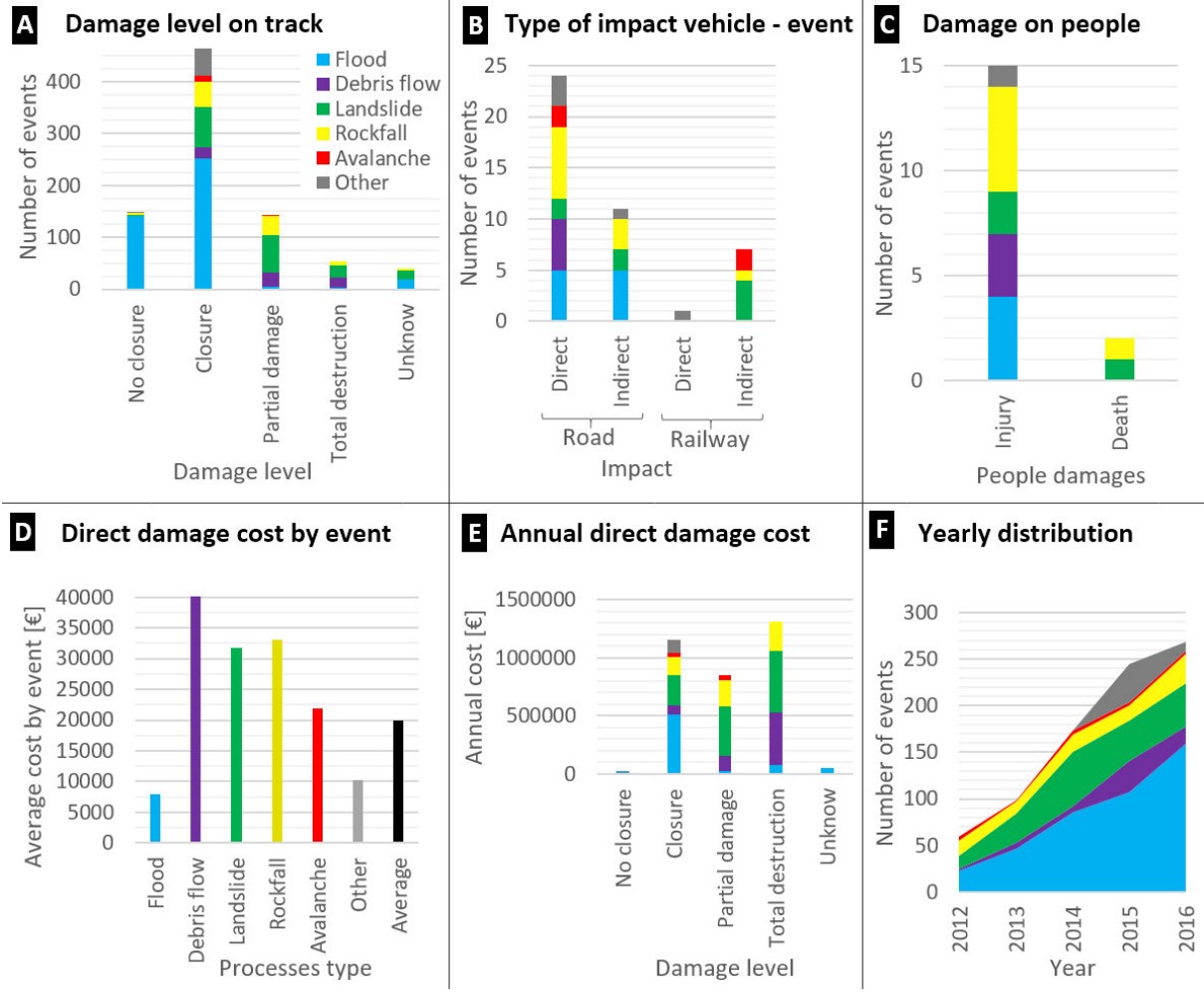

*Figure 6: A: Damage distribution. B: Distribution of impact types for vehicles on roads or railways and natural hazard events. C: Distribution of injuries and deaths. D: Distribution of the average event direct cost. E: Distribution of the annual direct cost. F: Annual distribution.*

4.5.2   To vehicles

5% (43 events) of all collected events generated damages to vehicles (Figure 6B and Table 16-SM). 3% (25) of events included direct impacts on vehicles and 2% (18) caused indirect impacts on vehicles (when a vehicle collides with material on the track). Except for a falling tree, which affected a tram directly, all direct impacts concerned roads. Two trains were affected indirectly by avalanches, four trains by landslides and one train by rockfalls. Only 1% (1 event) of events affecting railways caused a direct impact whereas 7% (7) of events caused indirect impacts. Conversely, 3% (24) of events affecting roads generated direct impacts and 1% (11) caused indirect impacts.

4.5.3   To people

People are rarely directly affected by events. 98.2% (831 events) of events did not cause injuries and 1.8% (15 events: 13 on roads and 2 on rail tracks) caused injuries (Figure 6C and Table 17-SM). 5.2% (5) and 4.3% (3) of events resulted in injuries; rockfalls and debris flows

generated the highest percentage of injuries. Twenty injured persons were identified, 10 of
which were in a train derailment in the Canton of Grisons due to a landslide in August 2014.
Two events (0.2%) caused death: the abovementioned event in Grison and an event where a
coach without passengers was directly impacted by a rockfall, killing the driver instantly in
March 2012 in Grisons. Only 0.1% (1) of events on roads caused death and 1% (1) of events
killed people on railways.
4.5.4    Closure duration
The closure duration for 35% of events (296 events) was collected from online press articles.
Half of the closures (148) lasted less than one day, and 41% (121) lasted one day to one week.
9% (27) of events lasted over one week, with a maximum of 15 months (Figure 5D). Thus,
87% (65) of floods induced closure durations of one day or less. This percentage decreased to
71% (5) for avalanches, 62% (36) for rockfalls, 59% (65) for landslides and 37% (15) for
debris flows.
4.5.5    Deviation length for roads
For three quarters (638 events) of the cases in which a deviation was possible, the lengths
varied from 1 km to 350 km (Figure 5E and Table 18-SM). Forty percent (255) of all
deviation track lengths were 1 km or less. One quarter (159) of deviation lengths were 2 to 9
km, 16% (100) of lengths were 10 to 19 km and the remaining 19% (124) of deviation paths
were over 20 km. The average deviation length was 40 km in the Alps, 9 km in the Jura and 7
km in the Plateau.
4.5.6    Direct damage costs
Direct damage costs include all costs directly related to the repair of the track to ensure
normal traffic service, including the full repair costs of the tracks. They are difficult or almost
impossible to assess; however, direct damage costs are important to determine an order of
magnitude of the costs that are directly induced after a natural hazard event affecting a
transportation track.
From 2012-2016, the annual direct damage costs for Swiss transportation track was estimated
at EUR 3.4 million. For one event, the average direct cost was EUR 19 900. On average, it
was EUR 8000 for floods, EUR 47 800 for debris flows, EUR 31700 for landslides, EUR 33
100 for rockfalls, EUR 21 900 for avalanches and EUR 10 200 for "other" events (Figure 6D
and Table 19-SM). The annual costs correspond to EUR 1.3 million for "total destruction",
EUR 1.2 million for "closure" and EUR 0.8 million for "partial damage" (Figure 6E). On
average, a "small" event costed EUR 15 800 and "medium" and a "large" events costed EUR
76 200 and EUR 175 700, respectively.
Small events (95% of all events; 804 events) represented 76% (2.6 mio EUR) of the total
direct damage costs, middle events (4%; 33) represented 15% (0.5 mio EUR) of the costs, and
large events (1%; 9) represented 9% (0.3 mio EUR) of the costs. Roads (93% of the total
transportation network length) represented 73% (2.5 mio EUR) of the total cost and railway
tracks (7% of all Swiss tracks) represented 27% (0.9 mio EUR) of all direct damage costs.

## 434  5  Discussion

### 435  5.1  Completeness of the database

The quality of the presented database is affected by several factors. The online press articles,
the main source of this database, did not report all natural hazard events affecting the Swiss
transportation network. This is particularly the case for events of small intensity. The
reporting of such events in articles depends on the number of casualties, the severity of the
injuries, the resources available for creation of the article, the preventive or educational
interest, and the presence of images. Article occurrence was theoretically higher in summer,
when the news activity is lower because of quieter political activity. In some cases, the
sensitivity increased, for example, after two tourists were killed on Gotthard highway in 2006
when a portion of the Eiger summit collapsed. This made journalists prone to focusing on
slope mass movements (RTS, 2006a and 2006b; Liniger and Bieri, 2006; Oppikofer et al.,
2008). Conversely, when many events occur simultaneously during intense storms, only the
most significant disasters are reported in the press. The event reporting likely depends on the
perception linked to the region of occurrence and the type of transportation network. For
instance, a 0.5 m$^3$ rockfall on a railway track in the Plateau has more media impact than one
occurring on an alpine road, where such events are more common and the consequences on
the traffic are lower.
The events collected from 2012-2016 ranged from 60 to 269 events per year (Figure 6F and
Table 20-SM). This may be biased because Google Alerts were only used after May 2014.
The data collection was less systematic for 2012 and 2013, with 60 and 99 events,
respectively. With Google Alerts, the number increased to 245 and 269 for 2015 and 2016,
respectively. With 173 events, 2014 was a transitional year, with Google Alerts used for
approximately half of the year. An advantage of Google Alerts is the variety of the online
sources from almost all the available online newspapers, which is better than the single source
(Badoux et al., 2016). Google Alerts allows for improving the event collection for floods.
Moreover, the total number of events increased yearly, even after the use of Google Alerts,
due to the increase in flood disruptions (Figure 6F). This shows that the use of Google Alerts
is not fully responsible for the yearly increase in the number of events. These numbers depend
strongly on the weather conditions that vary yearly. This demonstrates that the event
distribution is strongly dependent on a limited number of meteorological events such as long
rainfalls or severe storms.
Statistical predictions regarding a small sample of events are intrinsically imprecise (Davies
2013). The annual cost of damages from natural hazards in Switzerland (Hilker, 2009) from
1972-2007 shows great damage disparities over the years because extreme rainfall events or
successive storms greatly increase the number of events in one year.
From a geographic point of view, the collected data should be considered a snapshot of a short
time period capturing the background of "small" intensity events, representing 96% of the
total events and 76% of the total direct damage costs.
Notably, a number of natural hazard events induce expensive maintenance operations without
affecting the traffic, for example, by damaging protective infrastructure. Those events are not
considered in this study because they do not generate traffic perturbation but they should be
considered in risk management.
5.2   Event definition
The terminology of natural hazard events on roads and railways is partially inappropriate
because, although the origin of the direct event is typically natural (e.g., rainfall), the indirect
origin is often anthropic. The construction of a transportation network, its use, and
maintenance induce severe changes or actions that potentially affect slope stability, according
to the Terzaghi (1950) classification of the mechanism of landslides (Jaboyedoff et al.,
2016a). These causes of destabilizations, such as slope re-profiling, groundwater flow
perturbation, surface water overland flow modifications, land degradation, inappropriate
artificial structures, traffic vibration and ageing of infrastructure affect the landslide
occurrence (Larsen and Parks, 1997; Jaboyedoff et al, 2016). Furthermore, new infrastructure
around tracks often induces an under-sizing of the existing drainage systems, which can
induce the concentration of the surface or ground water flow and destabilize slopes. People
are thereby very often responsible for aggravation of the hazard consequences for built areas
without having sufficient knowledge of the natural hazards and associated risk. Laimer
(2017b) indicated that, along Austrian railways, 72% of events are human-induced.
## 5.3  Event trends
Minor and medium-sized natural hazard events are not well documented because their direct
consequences are often rapidly fixed, i.e., when the road can be re-opened within a few hours
of the event or is only partially closed.
The slope angle values are lower than common values for natural hazard slopes because they
are not the slope angles at the event origin but at the end of the propagation, as tracks are
generally located much lower than the sources of propagation.
Several factors must be considered in the slope distribution. One explanation for the lower
number of events on north-facing slopes is that there are fewer tracks due to the lower number
of buildings on these slopes. Furthermore, north-oriented slopes receive less solar heat than
south-oriented slopes and thus have fewer freeze-thaw cycles. This can partially explain the
high number of rockfalls on west, south and east-oriented slopes.
The monthly distribution indicates that floods mostly depend on two meteorological
conditions: thunderstorms and long-lasting rainfalls, which mainly occur in spring and
summer, particularly in combination with snowmelt in summer. The near absence of floods in
winter is the result of the Swiss winter climate, with a lack of long or brief but intense
precipitations and precipitation in mountains falling as snow. However, exceptions are
possible, such as floods caused by winter storms in January 2018 (RTS, 2018). Debris flows
mostly occurred in summer as the result of powerful and stationary thunderstorms. Landslides
mainly occurred in spring due to long-lasting rainfalls with the melting snow, generating
water, saturated soils and low evaporation. Snowmelt is the second trigger of landslides after
intense rainfalls on Austrian railway tracks for 2005-2015 (Laimer, 2017b). Laimer (2017b)
has shown that intense precipitation is a trigger for 78% of landslides on railway tracks in
Austria from 2005-2015. Freeze-thaw cycles during the winter are also a strong trigger of
rockfalls.
Rockfalls do not follow the trend of occurring mainly in spring and summer. They occur in
every season, mainly in autumn, winter and spring due to numerous freeze-thaw cycles during
these seasons, which weaken the cohesion of rocks. Unsurprisingly, avalanches occurred
mostly in winter. They occurred also in autumn as the result of fresh avalanches on soils that

are not yet covered with snow and non-effective winter track closures of roads in the Alps. The absence of avalanches in the spring is likely due to the presence of road winter closures.

Floods mostly occurred in the afternoon, probably after strong thunderstorms. Debris flows mostly occurred in the evening, probably after strong thunderstorms in the late afternoon or in the early evening. Landslide event triggers were not time dependent as the other event processes were. Rockfalls appear to be triggered during thawing, which occurs mostly in the morning. Snowdrifts from the "other" category began in the afternoon, after a few hours of strong wind. This is why the "other" category events are concentrated in the afternoon. Notably, the time of the event does not always match the actual event time, especially for events occurring during the night or on tracks with little traffic such as country roads.

The high proportion of landslides on train tracks can be explained by the presence of soil embankments or unsuitable filled material along railway tracks and due to their inclination limitations. In addition, despite more protections than average, highways are proportionally more vulnerable than other roads because of the alignment with many imposing cuts and fills. Similar to motorways, railway tracks require a balanced gradient ratio and thus must run along valley sides over far distances. This requires long and steep cut slopes (Laimer, 2017b).

Regional railway tracks may suffer from a lack of maintenance on track embankments during recent decades, which caused landslides and rockfalls on old age infrastructures that were built long before the basics of soil mechanics were understood (Terzaghi, 1925; Michoud et al., 2011; Laimer 2017a, 2017b).

The higher number of direct impacts (24) than indirect (11) impacts on roads shows that drivers can generally stop their vehicles before being affected by a fallen event unlike trains, which cannot be stopped within a short distance and reach the fallen mass (7 indirect impacts and one direct impact). In addition, there is a much higher probability that a vehicle on a road would be directly impacted by an event than a train on a track because the road traffic is excessively denser than the railway traffic.

Deviation lengths for railways are difficult to evaluate. In the case of replacement buses, the distance of deviation is calculated using the distance of the replacement buses on the road. For 72 events on railways (75% of all events on train tracks), there were no possibilities of deviations using other train tracks. In cases of no replacement service, the deviation length for the railway was the distance of train track between the two stations on both sides of the track closure. The average distance of deviation for this configuration was 65 km.

An example of an event from our database can be summarized as follows: a flood event
occurred in June during afternoon in the Plateau region on a small south-oriented slope with a
minor road. It generated a road closure of several hours with a deviation distance of less than
one kilometre and caused no injuries or deaths. The possibility of road deviation is large. On
the day of the event, the sun shined for half of the day, 10 mm of rain fell (20 mm during the
previous 5 days and 35 mm during the last 10 days) and the average temperature during the
event was 20°C. There were approximately 1 000 lightings around the event location on the
event day and the wind speed was 7 km/h in a north-east direction.
## 5.4  Direct damage cost estimation
Direct damage costs include all costs directly related to the rehabilitation of the track to
ensure traffic service. All repair costs of the tracks are included. The estimated direct costs did
not consider indirect costs such as vehicle repairs (the repair of a train costs a lot),
implementation of deviations, replacement buses in case of railway closure, costs generated
due to the traffic restriction for road and railway users or mitigation work and protective
measures.
The estimation of direct damage costs depends on many factors that are difficult to estimate.
The hour has an impact on the cost: repair work during the night or the weekend cost more
than those during office hours. The event location also affects the costs, for example, costs in
an alpine valley far from construction companies are higher than those in an agglomeration
where construction machines and landfill for the excavated material are nearly. The date also
impacts the costs: an event occurring during a period where weather conditions are difficult
will last longer. The emergency of the situation also influences the direct costs, as damage on
a secondary road or a highway will be treated with a different emergency level. There were
also influences from traffic, the presence of damaged retaining walls and protective measures,
the slope angle, the financial situation of the administration responsible for the repair work,
and the necessity of work on the slope or cliff above the track. Work on railways costs more
than that on roads because the access is often more difficult and because contact lines and rail
repairs can be more expensive.
An estimation of the direct costs of the "small" events is more credible than the costs of
events of higher damages because the main work is to clear the road of fallen materials. Cost
estimation for the "middle" and "larges" events is more complicated because the repairs
require large construction sites, which have their own characteristics that cannot be
generalized.
The estimated costs must be considered as an order of magnitude of the direct costs generated
by natural hazard events on the Swiss transportation network. These costs could be up to 10
times higher than the given cost estimation. However, the results are more refined than those
of the previous study of Voumard et al. (2016), where costs of events below EUR 8500 were
not considered.
Compared to the annual direct damage cost estimation of EUR 3.4 million for natural hazards
on the Swiss transportation network, annual damages caused by natural disasters in
Switzerland for 1972-2011 are estimated at EUR 290 million per year (OFEV, 2013).
Switzerland allocates EUR 2.5 billion each year for protection against natural hazards, which
corresponds to 0.6% of its GDP. 21% (EUR 0.5 billion) of this allocated amount concerns
intervention and repair (OFEV/OFS, 2007; OFEV/OFS, 2011).
5.5  General discussion of natural hazards and transportation networks
There are several methods to quantify the costs of track closures (Nicholson, 1997; Erath
2009). However, they are unsatisfactory because the quantification of costs, especially the
indirect costs, is difficult and the resilience must be carefully considered, as people often find
solutions to bypass the track closure (deferred travel, meeting realized with digital
technologies, alternative sources of supply, etc.).
The closure costs due to natural hazards, such as traffic congestion costs, are not compensated
for in Switzerland. However, models must include the potential loss of income in taxes if the
economy of the region is slowed. In addition, there are several ways to replace a
transportation route or means. For example, trains can be replaced by buses between two
stations. Using other train routes can be very complicated and long. Road deviation is usually
much easier; however, in some valleys in the Alps, the deviation lengths can reach hundreds
of kilometres and there may be no possibility of deviation. Notably, the increase of the travel
duration in the case of railway closures is more relevant for passengers than the distance of
deviation.
The spatial distribution (Figure 2) indicates a high density of events in populated areas,
principally on the Plateau. This concentration of events around populated areas can be
explained by various factors. First, when a meteorological event occurs in a densely populated
area, it may primarily affect tracks because the transportation networks are dense in those

areas. Conversely, a meteorological event that covers a similar surface but occurs in a sparsely populated area, for example, in an alpine lateral valley, will affect few tracks. Second, the number of people impacted, the associated economic consequences, the population sensitivity, the number of journalists available and the number of reporter-readers impact the media coverage of the natural hazard events. This leads to better media coverage of events in densely populated areas.

Davies (2013) notes the importance of the event in the context of the affected persons. A minor landslide that affects a person is unworthy of notice to the vast majority of the population but is considered momentarily catastrophic for the person, as it must reconsider its travel, find an alternative route or cancel its appointment.

Information acquisition is challenging in the development of such a database because it depends on several people working in the field, such as road menders, railway maintenance workers and forestry workers, who may have little time or interest in filling in the relevant attributes of the database. Hence, improvements to the database quality are possible using new tools such as off-line collaborative web-GIS (Balram, 2006; Pirotti et al., 2011; Aye et al. 2016; Olyazadeh et al., 2017), which can facilitate event data collection directly in the field using smart phones.

Furthermore, data acquisition and data analysis should distinguish the specific types of transportation networks. For instance, the sensibility to a natural hazard event on a railway track, where a $1 \text{ dm}^3$ rock can derail a train, is different from the sensitivity of an alpine road to the same volume of rock. Similarly, a landslide generating a track gauge change of 1 cm can lead to a train derailment whereas a landslide inducing track displacement of few tens of centimetres will probably not seriously affect the traffic of a mountain road. The liabilities in case of accidents on a railway track or road also differ. The railway manager and operator are responsible for the passengers' safety whereas the road manager allocates part of the responsibility to the driver. Therefore, compared to the road network, the railway network has a much higher sensitivity. The collection of the natural hazard events affecting roads and railways can be improved using different communication channels including social media such as the Facebook page of the Colorado Department of Transport (CDT) in the United States. This diffusion channel allows for the CDT to highlight natural hazard events that affect roads in Colorado department, informing drivers of their travel impacts.

# 6 Conclusions and perspectives

Using newspapers and Google Alerts, 846 natural hazard events that affected the Swiss transportation network from 2012 to 2016 were collected. They were characterized by 172 attributes, making them unique to Switzerland (Table 1). Our results highlight the impact of natural hazards on Swiss roads and railways, especially for small events with material deposits of less than 10 $m^3$ on the track that are rarely collected. They represent 95% of events in the database. The direct costs of all events were estimated at EUR 3.4 million per year with an average cost at EUR 19 900 per event. The direct costs of small events were estimated at EUR 2.5 million per year, which represents three quarters of the total direct costs.

Because of the increase in extreme meteorological events such as severe storms, climate change, rapidly growing infrastructure, increased traffic and the lack of funding for track maintenance, we expect increasing impacts of natural hazards on Swiss transportation networks. The key to reducing the natural hazard risk on tracks is financing.

The presented database and its event analysis can aid decision makers at the three Swiss political levels (the Confederation, the cantons and the municipalities) to plan and enforce protective measures in case of observable hot spots in the database.

Risk management in Switzerland may be improved by the existence of such a database. For example, it shows the important alternative ways to bypass obstacles. We highlighted that there were no deviation routes for one quarter of events. This proportion is high and must be considered by the authorities. The protection of all Swiss tracks against natural hazard processes would be too expensive. Thus, it is essential to ensure alternative tracks and fund protective measures according to the best ratio (cost/risk reduction). Minor roads often belong to the municipalities, which do not have a great interest in maintaining them. The Cantons and the Confederation would be advised to participate in or take over the maintenance of some roads that can be vital during the closure of main roads or railway tracks. This is particularly appropriate in the transportation corridor, where the minor road is located on the opposite side of the valley from the major road. This database aids in understanding the risk of transportation networks at the national scale rather than a track scale.

For this purpose, we created open access online maps of the events in Google Maps and ArcGIS Online (Figure 5-SM-AA and Figure 6-SM-AA) to promote this problematic issue. Our analysis also helps to elucidate the impacts of low-intensity events that had been considered almost insignificant and were largely unrecognized.

## Data availability

The data used in this paper are available upon request.

## Competing interests

The authors declare that they have no conflicts of interest.

## Acknowledgements

The authors would like to thank Y. Christen (Mantegani & Wysseier Ingenieure & Planer AG, Biel/Bienne, Switzerland) for their help in estimating direct remediation costs. We are also very grateful to the reviewers, especially to H.-J. Laimer (ÖBB, Salzburg, Austria) for their reviews and valuable comments. The authors are also thankful to N. Pollet (ALTAMETRIS, Paris, France) for providing meaningful comments about the data mining differences in railway and road networks. We also thank Z. C. Aye for her contribution in proofreading the manuscript.

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

## Supplementary Material

*Table 1-SM: 51 key words (in red) used in the Google Alerts to create the database. The numbers between brackets in the following tables refer to the number of elements considered according to the line or column attribute.*

| English | French | German | Italian |
|---|---|---|---|
| avalanche | avalanche | Lawinne | valanga |
| bad weather | intempéries | Unwetter | |
| flood | | Hochwasser | |
| hail | grêle | Hagel | |
| heavy rainfall | forte pluies | Heftige Regen | |
| ice avalanche | | Eislawine | |
| inundation | | Überflutung | |
| inundation | inondation | Überschwemmung | |
| landslide | glissement de terrain | Erdrutsch | frana |
| landslide | | Hangrutsch | |
| landslide | | Hachrutsche | |
| landslide | | Rüfenniedergang | |
| landslip | glissement | Rutschung | |
| mountain | pan de montagne | | |
| mud | boue | Schlamm | |
| mudflow | coulée de boue | Schlammlawine | |
| mudslide | | Erdlawine | |
| pirock | caillou | Stein | massi |
| rockfall | | Bergsturz | |
| rockfall | | Felsabbruch | |
| rockfall | éboulement | Felsbrock | |
| rockfall | écroulement | Felsbrocken | |
| rockfall | | Felssturz | |
| rockslide | chute de blocs | Steinschlag | cadono sassi |
| scree | | Geröll | |
| scree | éboulis | Schutt | |
| storm | tempête | Sturm | |
| thunderstorm | orage | Gewitter | |
| under water | sous l'eau | | |
| wind | vent | Wind | |

*Table 2-SM: Cost value estimation by square metre for the cost evaluation according to event importance, damage level and transport mode.*

| Damage level [EUR] | Cost per m$^2$, small event, road | Cost per m$^2$, middle event, road | Cost per m$^2$, large event, road | Cost per m$^2$, small event, train | Cost per m$^2$, middle event, train | Cost per m$^2$, large event, train |
|---|---|---|---|---|---|---|
| No closure | 5 | 5 | 5 | 5 | 5 | 5 |
| Closure | 85 | 130 | 170 | 300 | 340 | 385 |
| Partial damage | 255 | 300 | 340 | 470 | 510 | 555 |
| Total destruction | 850 | 890 | 980 | 1065 | 1105 | 1145 |
| Unknown damage | 130 | 170 | 215 | 255 | 300 | 340 |

*Table 3-SM: Distribution of event locations by Swiss geomorphologic-climatic region and event process.*

| Geomorphologic-climatic region | Flood (420) | Debris flow (69) | Landslide (192) | Rockfall (96) | Avalanche (16) | Other (53) | Average |
|---|---|---|---|---|---|---|---|
| Jura (98) | 19% | 0% | 3% | 6% | 0% | 15% | 12% |
| Plateau (371) | 57% | 4% | 42% | 6% | 0% | 79% | 44% |
| Alps (377) | 24% | 96% | 55% | 88% | 100% | 6% | 44% |
| Total (846) | 100% | 100% | 100% | 100% | 100% | 100% | 100% |

*Table 4-SM: Distribution of event locations by event process.*

| Event location | Flood (420) | Debris flow (69) | Landslide (192) | Rockfall (96) | Avalanche (16) | Other (53) | Average |
|---|---|---|---|---|---|---|---|
| Town (151) | 15% | 0% | 9% | 1% | 0% | 6% | 18% |
| Village (261) | 46% | 14% | 12% | 6% | 13% | 4% | 31% |
| Forest (185) | 4% | 46% | 38% | 58% | 13% | 13% | 22% |
| Unforested (249) | 0% | 6% | 5% | 12% | 69% | 0% | 29% |
| Total (846) | 100% | 100% | 100% | 100% | 100% | 100% | 100% |

*Table 5-SM: Distribution of slope angle by event process.*

| Slope angle | Flood (420) | Debris flow (69) | Landslide (192) | Rockfall (96) | Avalanche (16) | Other (53) | Average |
|---|---|---|---|---|---|---|---|
| 0°-10° (339) | 62% | 17% | 12% | 5% | 6% | 68% | 40% |
| 10°-20° (257) | 31% | 43% | 29% | 19% | 38% | 28% | 30% |
| 20°-30° (131) | 4% | 23% | 33% | 31% | 38% | 2% | 15% |
| 30°-40° (85) | 2% | 12% | 21% | 26% | 19% | 0% | 10% |
| 40°-50° (26) | 0% | 4% | 4% | 14% | 0% | 2% | 3% |
| 50°-60° (6) | 0% | 0% | 1% | 4% | 0% | 0% | 1% |
| 60 and higher (2) | 0% | 0% | 1% | 1% | 0% | 0% | 0% |
| Total (846) | 100% | 100% | 100% | 100% | 100% | 100% | 100% |

*Table 6-SM: Distribution of event importance by event process.*

| Location of process origin | Flood (420) | Debris flow (69) | Landslide (192) | Rockfall (96) | Avalanche (16) | Other (53) | Average |
|---|---|---|---|---|---|---|---|
| Small[1] (804) | 100% | 78% | 96% | 24% | 81% | 100% | 95% |
| Middle[2] (33) | 0% | 19% | 3% | 43% | 19% | 0% | 4% |
| Large[3] (9) | 0% | 3% | 1% | 33% | 0% | 0% | 1% |
| Total (846) | 100% | 100% | 100% | 100% | 100% | 100% | 100% |

[1] Small event: volume of deposit material on the track <10 m$^3$.
[2] Middle event: volume of deposit material on the track of 10-2000 m$^3$.
[3] Large event: volume of deposit material on the track > 2000 m$^3$.
*Table 7-SM: Distribution of the distance of the process origin by event process.*

| Distance of the process origin | Debris flow (69) | Landslide (192) | Rockfall (96) | Avalanche (16) | Other (53) | Average |
|---|---|---|---|---|---|---|
| Near[1] (185) | 0% | 52% | 33% | 6% | 100% | 35% |
| Far[2] (146) | 100% | 11% | 43% | 94% | 0% | 39% |
| Unknown (95) | 0% | 37% | 24% | 0% | 0% | 26% |
| Total (426) | 100% | 100% | 100% | 100% | 100% | 100% |

[1] Near: 0-50 m from the track.
[2] Far: > 50 m from the track.
*Table 8-SM: Distribution of the location of the process origin by event process.*

| Location of process origin | Debris flow (69) | Landslide (192) | Rockfall (96) | Avalanche (16) | Other (53) | Average |
|---|---|---|---|---|---|---|
| Above track (339) | 100% | 60% | 89% | 100% | 100% | 80% |
| Below track (29) | 0% | 14% | 2% | 0% | 0% | 7% |
| Unknown (58) | 0% | 26% | 9% | 0% | 0% | 14% |
| Total (426) | 100% | 100% | 100% | 100% | 100% | 100% |


*Table 9-SM: Rainfall [mm] during the natural hazard events.*

| Rainfall* [mm] | Flood | Debris flow | Landslide | Rockfall | Avalanche | Other | Average |
|---|---|---|---|---|---|---|---|
| Event day | 22 | 14 | 17 | 5 | 4 | 4 | 17 |
| Cum. last 5 days[1] | 49 | 32 | 57 | 27 | 32 | 15 | 45 |
| Cum. last 10 days[1] | 76 | 55 | 88 | 52 | 46 | 36 | 71 |
| Daily rain avg. last 5 days[2] | 10 | 6 | 11 | 6 | 6 | 3 | 9 |
| Daily rain avg. last 10 days[2] | 7 | 5 | 9 | 5 | 5 | 4 | 7 |
| Max daily rain last 5 days[3] | 30 | 21 | 32 | 15 | 18 | 11 | 27 |
| Max daily rain last 10 days[3] | 33 | 26 | 36 | 20 | 21 | 15 | 30 |
| Abs max daily rain[4] | 100 | 65 | 154 | 42 | 13 | 39 | - |
| Abs max daily rain last 5 days[4] | 154 | 75 | 154 | 77 | 140 | 39 | - |
| Abs max daily rain last 10 days[4] | 154 | 75 | 154 | 109 | 140 | 39 | - |

* Average by event process except for absolute values (last three lines of the table).
[1] Cumulative rainfall 5 and 10 days prior to the event day.
[2] Daily rainfall average 5 and 10 days prior to the event day.
[3] Maximum daily rainfall 5 and 10 days prior to the event day.
[4] Absolute maximum rainfall recorded (i.e., for one event) on the event day, 5 and 10 days prior to the event day.

*Table 10-SM: Monthly distribution of events by event process.*

| Year | Flood (420) | Debris flow (69) | Landslide (192) | Rockfall (96) | Avalanche (16) | Other (53) | Average |
|---|---|---|---|---|---|---|---|
| January (27) | 0% | 4% | 4% | 15% | 6% | 0% | 3% |
| February (65) | 0% | 1% | 6% | 6% | 19% | 81% | 8% |
| March (26) | 1% | 0% | 2% | 13% | 50% | 2% | 3% |
| April (28) | 2% | 0% | 6% | 7% | 0% | 2% | 3% |
| May (107) | 13% | 10% | 16% | 15% | 0% | 2% | 13% |
| June (253) | 41% | 16% | 29% | 7% | 0% | 8% | 30% |
| July (210) | 31% | 51% | 19% | 8% | 0% | 2% | 25% |
| August (35) | 4% | 12% | 4% | 1% | 0% | 2% | 4% |
| September (14) | 1% | 6% | 2% | 2% | 0% | 0% | 2% |
| October (14) | 1% | 0% | 1% | 10% | 0% | 0% | 2% |
| November (58) | 6% | 0% | 9% | 11% | 6% | 2% | 7% |
| December (9) | 0% | 0% | 1% | 4% | 19% | 0% | 1% |
| Total (846) | 100% | 100% | 100% | 100% | 100% | 100% | 100% |

*Table 11-SM: Transport mode distribution by event process.*

| Transport mode | Flood (420) | Debris flow (69) | Landslide (192) | Rockfall (96) | Avalanche (16) | Other (53) | Total |
|---|---|---|---|---|---|---|---|
| Road (747) | 53% | 9% | 20% | 10% | 1% | 7% | 100% |
| Railway (99) | 27% | 2% | 42% | 20% | 4% | 5% | 100% |

*Table 12-SM: Road class distribution by event process.*

| Road class | Flood (393) | Debris flow (67) | Landslide (151) | Rockfall (76) | Avalanche (12) | Other (48) | Average |
|---|---|---|---|---|---|---|---|
| Highway (34) | 7% | 0% | 2% | 1% | 10% | 2% | 5% |
| Motorway (2) | 0% | 0% | 1% | 0% | 0% | 0% | 0% |
| Major transit road (99) | 11% | 8% | 11% | 36% | 36% | 6% | 13% |
| Regional road (94) | 11% | 7% | 18% | 18% | 9% | 8% | 12% |
| Urban road (426) | 65% | 37% | 48% | 38% | 36% | 82% | 57% |
| Minor road (72) | 4% | 42% | 15% | 4% | 9% | 2% | 10% |
| Forest or land trail (20) | 2% | 6% | 5% | 5% | 0% | 0% | 3% |
| Total (747) | 100% | 100% | 100% | 100% | 100% | 100% | 100% |



*Table 13-SM: Railway class distribution by event process.*

| Track class | Flood (27) | Debris flow (2) | Landslide (41) | Rockfall (20) | Avalanche (4) | Other (5) | Average |
|---|---|---|---|---|---|---|---|
| National (29) | 37% | 0% | 32% | 30% | 0% | 0% | 29% |
| Regional (66) | 56% | 100% | 68% | 70% | 100% | 60% | 67% |
| Tram (4) | 7% | 0% | 0% | 0% | 0% | 40% | 4% |
| Total (99) | 100% | 100% | 100% | 100% | 100% | 100% | 100% |


*Table 14-SM: Distribution of possibility of deviations by event process.*

| Possibility of deviation | Flood (420) | Debris flow (69) | Landslide (192) | Rockfall (96) | Avalanche (16) | Other (53) | Total |
|---|---|---|---|---|---|---|---|
| Large (342) | 63% | 17% | 15% | 8% | 0% | 52% | 40% |
| Middle (190) | 21% | 7% | 32% | 17% | 7% | 33% | 23% |
| Small (102) | 7% | 6% | 13% | 32% | 66% | 4% | 12% |
| No (212) | 9% | 70% | 40% | 43% | 27% | 11% | 25% |
| Total (846) | 100% | 100% | 100% | 100% | 100% | 100% | 100% |



*Table 15-SM: Distribution of track damage by event process.*

| Damage level | Flood (420) | Debris flow (69) | Landslide (192) | Rockfall (96) | Avalanche (16) | Other (53) | Total |
|---|---|---|---|---|---|---|---|
| No closure (149) | 34% | 0% | 1% | 3% | 6% | 4% | 18% |
| Closure (483) | 60% | 35% | 50% | 50% | 81% | 96% | 57% |
| Partial damage (143) | 1% | 39% | 37% | 39% | 13% | 0% | 17% |
| Total destruction (53) | 1% | 26% | 12% | 8% | 0% | 0% | 6% |
| Unknown damage (18) | 4% | 0% | 0% | 0% | 0% | 0% | 2% |
| Total (846) | 100% | 100% | 100% | 100% | 100% | 100% | 100% |


*Table 16-SM: Distribution of damage and impact on vehicles by event process.*

| Damage and impact type on vehicles | Flood (420) | Debris flow (69) | Landslide (192) | Rockfall (96) | Avalanche (16) | Other (53) | Total |
|---|---|---|---|---|---|---|---|
| No damage (803) | 98% | 93% | 96% | 89% | 80% | 89% | 95% |
| Vehicle damage: direct impact[1] (25) | 1% | 7% | 1% | 7% | 7% | 7% | 3% |
| Vehicle damage: indirect impact[2] (18) | 1% | 0% | 3% | 4% | 13% | 4% | 2% |
| Total (846) | 100% | 100% | 100% | 100% | 100% | 100% | 100% |

[1] Direct impact: a vehicle is directly affected by a hazard.
[2] Indirect impact: a vehicle collides with an event mass already fallen on the track.

*Table 17-SM: Distribution of injury and death by event process.*

| Injury and death | Flood (420) | Debris flow (69) | Landslide (192) | Rockfall (96) | Avalanche (16) | Other (53) | Total |
|---|---|---|---|---|---|---|---|
| No damage on people (828) | 99% | 96% | 98% | 93% | 100% | 98% | 98% |
| Injury (15) | 1% | 4% | 1% | 5% | 0% | 2% | 2% |
| Death (3) | 0% | 0% | 1% | 2% | 0% | 0% | 0% |
| Total (846) | 100% | 100% | 100% | 100% | 100% | 100% | 100% |


*Table 18-SM: Distribution of deviation length on roads by event process.*

| Deviation length | Flood (383) | Debris flow (21) | Landslide (116) | Rockfall (58) | Avalanche (11) | Other (49) | Mean |
|---|---|---|---|---|---|---|---|
| 0-1 km (255) | 58% | 29% | 12% | 9% | 0% | 12% | 40% |
| 2-5 km (102) | 14% | 38% | 16% | 3% | 0% | 39% | 16% |
| 6-9 km (57) | 9% | 10% | 9% | 7% | 0% | 14% | 9% |
| 10-19 km (100) | 9% | 5% | 34% | 21% | 0% | 22% | 16% |
| 20-49 km (63) | 5% | 0% | 17% | 26% | 45% | 8% | 10% |
| 50-99 km (24) | 3% | 5% | 5% | 12% | 0% | 0% | 4% |
| 100-249 km (30) | 2% | 14% | 6% | 17% | 18% | 4% | 5% |
| 250-350 km (7) | 0% | 0% | 0% | 5% | 36% | 0% | 1% |
| Total (638) | 100% | 100% | 100% | 100% | 100% | 100% | 100% |


*Table 19-SM: Direct damage cost distribution by events type.*

| Damage level [EUR] | Flood (420) | Debris flow (69) | Landslide (192) | Rockfall (96) | Avalanche (16) | Other (53) | Total |
|---|---|---|---|---|---|---|---|
| | | | Annual cost [EUR] | | | | |
| No closure (149) | 12 665 | 340 | 85 | 765 | 255 | 170 | 14 280 |
| Closure (483) | 514 250 | 71 400 | 262 650 | 160 650 | 28 900 | 107 950 | 1 145 800 |
| Partial damage (143) | 25 500 | 127 500 | 425 000 | 227 800 | 40 800 | 0 | 846 600 |
| Total destruction (53) | 72 250 | 459 850 | 528 700 | 246 500 | 0 | 0 | 1 307 300 |
| Unknown damage (18) | 45 900 | 0 | 0 | 0 | 0 | 0 | 45 900 |
| Annual cost [million €] | 0.67 | 0.66 | 1.22 | 0.64 | 0.07 | 0.11 | 3.36 |
| Avg. cost by event | 8 000 | 47 800 | 31 700 | 33 100 | 21 900 | 10 200 | 19 900 |


*Table 20-SM: Annual distribution of events by event process.*

| Year | Flood (420) | Debris flow (69) | Landslide (192) | Rockfall (96) | Avalanche (16) | Other (53) | Average |
|---|---|---|---|---|---|---|---|
| 2012 (60) | 5% | 3% | 7% | 17% | 25% | 2% | 7% |
| 2013 (99) | 11% | 10% | 16% | 14% | 6% | 2% | 12% |
| 2014 (173) | 20% | 10% | 30% | 20% | 25% | 0% | 20% |
| 2015 (245) | 25% | 49% | 22% | 17% | 25% | 77% | 29% |
| 2016 (269) | 38% | 28% | 24% | 33% | 19% | 19% | 32% |
| Total (846) | 100% | 100% | 100% | 100% | 100% | 100% | 100% |



*Table 21-SM: Summary of event process key features.*

| Attribute (with values of the greatest occurrence) | Flood | Debris flow | Landslide | Rockfall | Avalanche | Other | Mean |
|---|---|---|---|---|---|---|---|
| Event importance | Small | Small | Small | Small | Small | Small | Small |
| Yearly number of events | 84 | 14 | 38 | 19 | 3 | 11 | 169 |
| Months | 6, 7 | 7, 6 | 6, 7, 5 | 1, 5, 3, 11, 10 | 3 | 2 | 6, 7 |
| Season | Spring | Summer | Spring | Spring, Winter | Winter | Winter | Spring |
| Time of day | Afternoon | Afternoon | All day | All day | Morning | All day | Afternoon |
| Hour | 12-19 | 15-19 | 0-24 | 0-24 | 8-13 | 0-24 | 14-19 |
| Region | Plateau | Alps | Alps | Alps | Alps | Plateau | Alps, Plateau |
| Canton | Bern | Graubünden | Valais | Valais | Valais | Vaud | Bern |
| Slope angle | 0-10 | 10-20 | 20-30 | 20-30 | 10-20 | 0-10 | 0-10 |
| Slope orientation | S | W | S | W | N-W | S-E | S, S-W and W |
| Location | Village | Forest | Forest | Forest | Mountain | Country | Village |
| Damage on track | Closure | Partial dam. | Closure | Closure | Closure | Closure | Closure |
| Direct costs per event (Euro) | 6 900 | 39 000 | 25 700 | 261 000 | 155 000 | 8 600 | 16 000 |
| Track geometry | Str. line | Wide curve | Wide curve | Wide curve | Wide curve | S. line & w. curve | Wide curve |
| Crossing | Near | No | No | No | No | No | No |
| Closure duration | 3 hours | 1 week | 1 day | 3 hours | 1-2 days | 3 hours | 3 hours |
| Possibility of deviation | Large | No | No | No | Small | Middle | Large |
| Deviation length | 0-1 km | No deviation | No deviation | No deviation | 250-350 km | 2-5 km | 0-1 km |
| Event origin distance | - | Far | Near | Far | Far | Near | Near |
| Event above below | - | Up | Up | Up | Up | Up | Up |
| Altitude [m a.s.l.] | 525 | 1139 | 809 | 897 | 1274 | 614 | 701 |
| Track type | Road | Road | Road | Road | Road | Road | Road |
| Track importance | Minor | Minor | Minor | Minor | Minor | Minor | Minor |
| Rainfall event day [mm] | 22 | 14 | 171 | 5 | 4 | 4 | 17 |


*Figure 1-SM: Attributes of the database.*

| | EventID | Date — Number of attributes: 15 | | | | | | | | | | | | | | |
|---|---|---|---|---|---|---|---|---|---|---|---|---|---|---|---|---|
| **Category** | EventID | DATE | | | | | | | | | | | | | | |
| **Attribute** | EventID | D_IDdate | D_Year | D_Month | D_Day | D_MonthWeek | D_DayName | D_Season | D_Hour | D_HourPrecise | D_DayPart | D_IDDay | D_IDEventSameDay | D_SameClimLongPeriod | D_SameClimShortPeriod | MuenichRe |
| **Description** | Unique ID for each event | Unique ID for each event containing the date | Year of the event | Month of the event | Day of the event | Month divided into 4 quarters | Name of the day of the event | Season of the event | Hour of the event hourly rounded | Hour of the event | Day part of the event | Unique ID for each event day (same ID when >1 event per day) | Unique ID for event same day | Long time period in which the event is included | Short time period in which the event is included | Period given by MünichRe in which the event is included |
| **Unit** | - | y m d XX | year | month | day | - | - | - | h:m:s | h:m:s | - | y m d | - | y.m.d-y.m.d | y.m.d-y.m.d | y.m.d-y.m.d |
| **Exemple** | 431 | 2015050400 | 2015 | 5 | 4 | 5-1 | Monday | Spring | 10:00:00 | 10:15:00 | Morning | 20150504 | 2 | 2015.04.27-2015.07.25 | 2015.04.27-2015.05.07 | 2014.06.03-2014.06.12 |
| **Comment** | - | - | From 2011 to 2015 | - | - | First quarter (1) of the 5th month (5) | Useful to categorise business day and weekend | - | - | - | 5 parts: morning, afternoon, evening, night and unknown | Allow to recognise the day when with several events | The maximal ID by event day gives the nb of events during this day | | | From MuenichRe yearly natural catastrophes analysis |
| **Source** | - | - | Online article | Online article | Online article | Online article | Online article | Online article | Online article | Online article | Online article | - | - | - | - | MünichRe |
| | 1 | 2 | 3 | 4 | 5 | 6 | 7 | 8 | 9 | 10 | 11 | 12 | 13 | 14 | 15 | 16 |

| | Location — Number of attributes: 21 | | | | | | | | | |
|---|---|---|---|---|---|---|---|---|---|---|
| **Category** | | | | | | | | | | |
| **Attribute** | L_Canton | L_Commune | L_Detail | L_Precision | L_SitGeo | L_OriSlope | L_Urbanity | L_Slope | L_SlopeRound | L_Lanscape |
| **Description** | Canton where occurs the event | Commune where occurs the event | Detail to help the location | Precision of the location | Geographical situation of the event | If slope: orientation of the slope | Urbanity of the event | Slope angle average in an 25 meter radius around the event | Slope angle rounded to the nearest ten | Lanscape of the event locaiotn |
| **Unit** | - | - | - | - | - | - | - | [°] | [°] | |
| **Exemple** | Valais | Bagnes | - | Accurate | Slope | North-East | Forest | 13 | 13 | Dry mountainous landscape of western central Alps |
| **Comment** | - | - | - | Three levels of accuracy: accurate, middle and communal accuracy | Four classes: plain, ridge, slope and valley bottom | Nine classes: north, north-east, south-east, south, south-west, west, noth-west and any slope | Seven classes: mountain, forest, country, hamlet, village, agglomeration and town | From 0 ° to 56° | From 0 ° to 60° | 36 types |
| **Source** | Online article | Online article | Online article | Online article and map | Map | Map | Map | GIS | GIS | GIS |
| | 17 | 18 | 19 | 20 | 21 | 22 | 23 | 24 | 25 | 26 |

| LOCATION | | | | | | | | | | |
|---|---|---|---|---|---|---|---|---|---|---|
| L_Areas | L_Area_reg | L_MN03_X | L_MN03_Y | L_MN03_Z | L_MN95_X | L_MN95_Y | L_MN95_Z | L_WGS84_Lo | L_WGS84_La | L_WGS84_Z |
| Areas of the event location | Regional area of the location | X coordinates in CH1903 coordinate system | Y coordinates in CH1903 coordinate system | Z coordinates in CH1903 coordinate system | X coordinates in CH1903+ coordinate system | Y coordinates in CH1903+ coordinate system | Z coordinates in CH1903+ coordinate system | Longitude in WGS84 coordinate system | Latitude in WGS84 coordinate system | ALtitude in WGS84 coordinate system |
| | | [m] | [m] | [m] | [m] | [m] | [m] | [°] | [°] | [m] |
| Alpine region | Alps | 588456 | 98247 | 1377 | 2588455 | 1098247 | 1377 | 7.289538659 | 46.03566307 | 1431 |
| 5 types: Alpine region, Swiss Plateau, Tabular Jura, Folded Jura and Independent | 3 types: Jura, Plateau and Alps | - | - | - | - | - | - | - | - | - |
| GIS | Map | GIS | GIS | GIS | GIS | GIS | GIS | GIS | GIS | GIS |
| 27 | 28 | 29 | 30 | 31 | 32 | 33 | 34 | 35 | 36 | 37 |

Event characterization    Number of attributes: 12

| Category | Event characterization | | | | | | | | | | | |
|---|---|---|---|---|---|---|---|---|---|---|---|---|
| Attribute | E_Type | E_TypePrec | E_UpDownst | E_UpDownst Risk | E_Provenan | E_Volume | E_Masse | E_Width | E_Importan | E_Other | E_PictureName | E_Picture |
| Description | Type of natural hazard event | Precise type of natural hazard event | Origin up or downstream of the natural hazard event | Origin up, downstream or only risk of the event | Estimation of the distance of the event origin | Volume of the event | Masse of the event | Width of the event mass on the track | Importance of the event | Other information | Picture name of the event | Picture |
| Unit | - | - | - | - | [m] or - | [m³] | [kg] | [m] | - | - | - | - |
| Exemple | Landslide | Landslide | - | - | - | - | - | - | Small | - | 2015050400.jpg | - |
| Comment | 6 types: rockfall, debris flow, landslide, avalanche, flood, other | 8 types: rockfall, debris flow, landslide, avalanche, flood, hail, snowdrift, falling tree | 3 classes: upstream, downstream and unknown | 4 classes: upstream, downstream, risk (no event, only preventive closure) and unknown | 3 classes: near (few meters to 10 meters, far (> 10 m) or prevention (only proventive closure) | Estimation of the falled volume on the track of the event | Masse of the event (only for rockfall) | - | 3 classes: small, middle, big (huge event) | - | - | - |
| Source | Online article | Online article | Online article | Online article | Online article | Online article | Online article | Online article | Online article | Online article | Online article | Online article or field visit |
| | 38 | 39 | 40 | 41 | 42 | 43 | 44 | 45 | 46 | 47 | 48 | 49 |


**Track caracterization**  Number of attributes: 17

| Category | | | | | | | Track caracterization | | | | | | | | | | |
|---|---|---|---|---|---|---|---|---|---|---|---|---|---|---|---|---|
| Attribute | T_Type | T_TrainClasses | T_RoadClasses | T_MajorMin | T_Closure | T_DetailClosure | T_ClosureDuration | T_ClosureDurationRound | T_Deviation | T_DistDev | T_DistDevRound | T_DevDetail | T_PossDevi | T_PopDirAf | T_PopIndAf | T_Sinuosity | T_crossing |
| Description | Distinction between road and railway | Classes of the affected train tracks | Classes of the affected road tracks | Simplified classification of track importance | Track closure or not | Detail of the track closure | Time of track closure in hours | Ronded time of track closure in hours | Deviation or not | Distance of the deviation path | Rounded distance of the deviation path | Deviation detail | Capacity to have other deviation paths | Population directly affected by the track closure | Population indirectly affected by the track closure | Sinuosity og the affected track | Crossing near of the event or not |
| Unit | - | - | - | - | - | - | [h] | [h] | - | [km] | [km] | - | - | - | - | - | - |
| Exemple | Road | White | White | Minor | Yes | - | 23 | 24 | - | 8 | 10 | - | Large | Any | Small | NSC | NO |
| Comment | 2 types: road or railwa | 3 classes: national, regional, tram | 8classes: highway, semi-highway, red, yellow, white, white dash and black | 2 classes: minor and major | Three classes: yes, no, unknown | - | - | - | 2 classes: yes or no | - | - | - | 4 classes: large, middle, small, any | 5 classes: very large, large, middle, small, any | 5 classes: very large, large, middle, small, any | 6 types: Straight Line, Wide Curve, Tight Curve, Near Wide Curve, Near Tight Curve | 4 types: IN a crossing, NEAR a crossing, NO crossinf in the area and unknown (not enough location accuravy |
| Source | Online article | Map | Map | Map | Online article | Online article | Online article | Online article | Map | Map | Map | Map | Map | Map | Map | Map | Map |
| | 50 | 51 | 52 | 53 | 54 | 55 | 56 | 57 | 58 | 59 | 60 | 61 | 62 | 63 | 64 | 65 | 66 |

**Damage**  Number of attributes: 11

| Category | | | | | Damage | | | | | | |
|---|---|---|---|---|---|---|---|---|---|---|---|
| Attribute | D_Form | D_Injured | D_InjuredNb | D_Death | D_DeathNb | D_Vehicle | D_ImpactTy | D_VehiType | D_VehiNb | D_TrackDetail | D_Infras_type |
| Description | Form of track damage | Injured people? | Number of injured people | Killed people? | Number of killed people | Damage to vehicle | Type of impact between vehicle and event | Type of damaged vehicle | Number of damaged vehicle | Detail of track damage | Type of instrastructure damage |
| Unit | - | - | - | - | - | - | - | - | - | | - |
| Exemple | ? | No | - | No | - | No | - | - | - | | - |
| Comment | 6 classes: ? (unknown), NC (no closure), C (closure due to sedimentation), P (partial damage), T (total destruction), and not studied | 2 types: yes or no | - | 2 types: yes or no | - | 2 types: yes or no | Three types: no impact, direct impact or indirect impact | - | - | | - |
| Source | Online article | Online article | Online article | Online article | Online article | Online article | Online article | Online article | Online article | | Online article |
| | 67 | 68 | 69 | 70 | 71 | 72 | 73 | 74 | 75 | 76 | 77 |


**Weather**  Number of attributes: 68

| Category | | | | | | | | | | | | | | | | | | | |
|---|---|---|---|---|---|---|---|---|---|---|---|---|---|---|---|---|---|---|---|
| Attribute | M_Meteo | M_Sun | M_Sun_avg_5d | M_Sun_avg_10d | M_Sun_max_5d | M_Sun_max_10d | M_Sun_min_5d | M_Sun_min_10d | M_Rain | M_Rain_5d_cum | M_Rain_10d_cum | M_Rain_max_daily_5d | M_Rain_max_daily_10d | M_Rain_avg_daily_5d | M_Rain_avg_daily_10d | M_Storm_near | M_Storm_near_sum_5d | M_Storm_near_sum_10d | M_Strom_near_max_daily_5d |
| Description | Rain information for a given time period | Percentage of sun during the event day | Percentage of sun of the last 5 days from event | Percentage of sun of the last 10 days from event | Maximum percentage of sun of the last 5 days from event | Maximum percentage of sun of the last 10 days from event | Minimum percentage of sun of the last 5 days from event | Maximum percentage of sun of the last 10 days from event | Rain the event day | Cumulative rain of the last 5 days from event | Cumulative rain of the last 10 days from event | Maximum daily rain of the last 5 days from event | Maximum daily rain of the last 10 days from event | Average daily rain of the last 5 days from event | Average daily rain of the last 10 days from event | Number of near storms the event day | Number of near storms of the 5 days from event | Number of near storms of the 10 days from event | Maximum daily number of near storms of the 5 days from event |
| Unit | | % | % | % | % | % | % | % | mm | mm | mm | mm | mm | mm | mm | - | - | - | - |
| Exemple | - | 4 | 29.4 | 34.1 | 77 | 98 | 0 | 0 | 0.2 | 28.7 | 38.4 | 19.9 | 19.9 | 5.74 | 3.84 | 0 | 0 | 0 | 0 |
| Comment | Only for som events | - | - | - | - | - | - | - | - | - | - | - | - | - | - | Near storm: <3 km around the weather station | Near storm: <3 km around the weather station | Near storm: <3 km around the weather station | Near storm: <3 km around the weather station |
| Source | Sturmarchiv | MeteoSwiss | MeteoSwiss | MeteoSwiss | MeteoSwiss | MeteoSwiss | MeteoSwiss | MeteoSwiss | MeteoSwiss | MeteoSwiss | MeteoSwiss | MeteoSwiss | MeteoSwiss | MeteoSwiss | MeteoSwiss | MeteoSwiss | MeteoSwiss | MeteoSwiss | MeteoSwiss |
| | 78 | 79 | 80 | 81 | 82 | 83 | 84 | 85 | 86 | 87 | 88 | 89 | 90 | 91 | 92 | 93 | 94 | 95 | 96 |

| M_Strom_near_max_daily_5d | M_Strom_near_max_daily_10d | M_Storm_far | M_Storm_far_sum_5d | M_Storm_far_sum_10d | M_Strom_far_max_daily_5d | M_Strom_far_max_daily_10d | M_Storm_all | M_Storm_all_sum_5d | M_Storm_all_sum_10d | M_Strom_all_max_daily_5d | M_Strom_all_max_daily_10d | M_Temp_min | M_Temp_min_5d | M_Temp_min_10d | M_Temp_max | M_Temp_max_5d | M_Temp_max_10d | M_Temp_avg | M_Temp_avg_5d |
|---|---|---|---|---|---|---|---|---|---|---|---|---|---|---|---|---|---|---|---|
| Maximum daily number of near storms of the 5 days from event | Maximum daily number of near storms of the 10 days from event | Number of far storms the event day | Number of far storms of the 5 days from event | Number of far storms of the 10 days from event | Maximum daily number of far storms of the 5 days from event | Maximum daily number of far storms of the 10 days from event | Number of all storms the event day | Number of all storms of the 5 days from event | Number of all storms of the 10 days from event | Maximum daily number of allstorms of the 5 days from event | Maximum daily number of all storms of the 10 days from event | Minimum temperature the event day | Minimum temperature the last 5 days from event | Minimum temperature the last 10 days from event | Maximum temperature the event day | Maximum temperature the last 5 days from event | Maximum temperature the last 10 days from event | Average temperature the event day | Average temperature the last 5 days from event |
| - | - | - | - | - | - | - | - | - | - | - | - | [°C] | [°C] | [°C] | [°C] | [°C] | [°C] | [°C] | [°C] |
| 0 | 0 | 0 | 0 | 2 | 0 | 1 | 2 | 3 | 10 | 1 | 5 | 7 | 1 | -3 | 14 | 14 | 15 | 10 | 7 |
| Near storm: <3 km around the weather station | Near storm: <3 km around the weather station | Far storm: >3 km around the weather station | Far storm: >3 km around the weather station | Far storm: >3 km around the weather station | Far storm: >3 km around the weather station | Far storm: >3 km around the weather station | - | - | - | - | - | - | - | - | - | - | - | - | - |
| MeteoSwiss | MeteoSwiss | MeteoSwiss | MeteoSwiss | MeteoSwiss | MeteoSwiss | MeteoSwiss | MeteoSwiss | MeteoSwiss | MeteoSwiss | MeteoSwiss | MeteoSwiss | MeteoSwiss | MeteoSwiss | MeteoSwiss | MeteoSwiss | MeteoSwiss | MeteoSwiss | MeteoSwiss | MeteoSwiss |
| 96 | 97 | 98 | 99 | 100 | 101 | 102 | 103 | 104 | 105 | 106 | 107 | 108 | 109 | 110 | 111 | 112 | 113 | 114 | 115 |


| M_Temp_avg_10d | M_Temp_min_Corr | M_Temp_min_5d_Corr | M_Temp_min_10d_Corr | M_Temp_max_Corr | M_Temp_max_5d_Corr | M_Temp_max_10d_Corr | M_Temp_avg_Corr | M_Temp_avg_5d_Corr | M_Temp_avg_10d_Corr | M_Temp_amp_Corr | M_Temp_amp_5d_Corr | M_Temp_amp_10d_Corr | M_Wind_avg |
|---|---|---|---|---|---|---|---|---|---|---|---|---|---|
| Average temperature the last 10 days from event | Corrected minimum temperature the event day | Corrected minimum temperature the last 5 days from event | Corrected minimum temperature the last 10 days from event | Corrected maximum temperature the event day | Corrected maximum temperature the last 5 days from event | Corrected maximum temperature the last 10 days from event | Corrected average temperature the event day | Corrected average temperature the last 5 days from event | Corrected average temperature the last 10 days from event | Ttemperature amplitude the event day | Temperature amplitude the last 10 days from the event | Temperature amplitude the last 5 days from the event | Average wind speed the event day |
| [°C] | [°C] | [°C] | [°C] | [°C] | [°C] | [°C] | [°C] | [°C] | [°C] | [°C] | [°C] | [°C] | [km/h] |
| 7 | 9 | 3 | -1 | 16 | 16 | 17 | 12 | 9 | 9 | 9 | 12 | 15 | 8 |
| - | Correction with height difference bewteen weather station and event location with lapse rate of -0.65 °C for + 100m altitude | Correction with height difference bewteen weather station and event location with lapse rate of -0.65 °C for + 100m altitude | Correction with height difference bewteen weather station and event location with lapse rate of -0.65 °C for + 100m altitude | Correction with height difference bewteen weather station and event location with lapse rate of -0.65 °C for + 100m altitude | Correction with height difference bewteen weather station and event location with lapse rate of -0.65 °C for + 100m altitude | Correction with height difference bewteen weather station and event location with lapse rate of -0.65 °C for + 100m altitude | Correction with height difference bewteen weather station and event location with lapse rate of -0.65 °C for + 100m altitude | Correction with height difference bewteen weather station and event location with lapse rate of -0.65 °C for + 100m altitude | Correction with height difference bewteen weather station and event location with lapse rate of -0.65 °C for + 100m altitude | - | - | - | - |
| MeteoSwiss | MeteoSwiss | MeteoSwiss | MeteoSwiss | MeteoSwiss | MeteoSwiss | MeteoSwiss | MeteoSwiss | MeteoSwiss | MeteoSwiss | MeteoSwiss | MeteoSwiss | MeteoSwiss | MeteoSwiss |
| 116 | 117 | 118 | 119 | 120 | 121 | 122 | 123 | 124 | 125 | 126 | 127 | 128 | 129 |

| M_Wind_avg_5d | M_Win_avg_10d | M_Wind_max | M_Wind_max_5d | M_Wind_max_10d | M_Wind_dir | M_Win_dir_5d | M_Win_dir_10d | M_Snow | M_Fresh_snow | M_Fresh_snow_5d | M_Fresh_snow_10d | M_Accronym_Stn_Weath | M_Alt_Stn_Weath | M_Diff_Alt_Stn_Weath_Event | M_Dist_Stn_Weath |
|---|---|---|---|---|---|---|---|---|---|---|---|---|---|---|---|
| Average wind speed the 5 last days from event | Average wind speed the last 10 days from event | Maximum wind speed the event day | Maximum wind speed the 5 last days from event | Maximum wind speed the last 10 days from event | Average wind direction the event day | Average wind direction the last 5 days from event | Average wind direction the last 10 days from event | Snow cover height the event day | Fresh snow cover height the event day | Fresh snow cover height the 5 last days from event | Fresh snow cover height the 5 last days from event | Accronym of the used weather station | Altitude of the used weather station | Altitude difference between the weather station and the even location | Distance between the weather station and the even location |
| [km/h] | [km/h] | [km/h] | [km/h] | [km/h] | [°] | [°] | [°] | [cm] | [cm] | [cm] | [cm] | - | [m] a.s.l. | [m] | [km] |
| 9 | 10 | 32 | 38 | 46 | 47 | 48 | 63.9 | 0 | 0 | 0 | 0 | ZER | 1638 | -261 | 36 |
| - | - | - | - | - | 0° = North, 90° = East, 180° = South, 270° = West | 0° = North, 90° = East, 180° = South, 270° = West | 0° = North, 90° = East, 180° = South, 270° = West | - | - | - | - | - | - | - | - |
| MeteoSwiss | MeteoSwiss | MeteoSwiss | MeteoSwiss | MeteoSwiss | MeteoSwiss | MeteoSwiss | MeteoSwiss | MeteoSwiss | MeteoSwiss | MeteoSwiss | MeteoSwiss | MeteoSwiss | MeteoSwiss | MeteoSwiss | MeteoSwiss |
| 130 | 131 | 132 | 133 | 134 | 135 | 136 | 137 | 138 | 139 | 140 | 141 | 142 | 143 | 144 | 145 |


**Geology** — Number of attributes: 11

| Category | Geology | | | | | | | | | | |
|---|---|---|---|---|---|---|---|---|---|---|---|
| Attribute | G_watershed | G_Geol | G_Tecto_f | G_Geol_f | G_Tec1_f | G_Tec2_f | G_Tec3_f | G_Acquifer | G_Hydrogeology | G_Productivity | G_Geology |
| Description | Watershed on the event | | | Geology | Tectonic 1 | Tectonic 2 | Tectonic 3 | Aquifer | Hydrogeology | Productivity of the event field | General geology |
| Unit | - | - | - | - | - | - | - | - | - | - | - |
| Exemple | RHONE | er | pi | Gneiss et micaschistes (y compris migmatites et phyllites; princ. metasediments) | Nappes de socle cristallin penniques moyennes | Nappe du Mont-Fort | - | Aquifer reservoirs in coherent rocks | Sparsely productive aquifer reservoirs in non-karstified, cracked and porous coherent rocks | Variable productivity | Sericite gneiss |
| Comment | - | - | - | - | - | - | - | - | - | - | - |
| Source | Swisstopo | Swisstopo | Swisstopo | Swisstopo | Swisstopo | Swisstopo | Swisstopo | Swisstopo | Swisstopo | Swisstopo | Swisstopo |
| | 146 | 147 | 148 | 149 | 150 | 151 | 152 | 153 | 154 | 155 | 156 |

**Source** — Number of attributes: 16

| Category | Source | | | | | | | | | | | | | | | |
|---|---|---|---|---|---|---|---|---|---|---|---|---|---|---|---|---|
| Attribute | Source1 | Source2 | Source3 | Source4 | Source5 | Source6 | Source7 | Source8 | Source9 | Source10 | Source11 | Source12 | Source13 | Source14 | Source15 | Source16 |
| Description | Source 1 for the event | Source 2 for the event | Source 3 for the event | Source 4 for the event | Source 5 for the event | Source 6 for the event | Source 7 for the event | Source 8 for the event | Source 9 for the event | Source 10 for the event | Source 11 for the event | Source 12 for the event | Source 13 for the event | Source 14 for the event | Source 15 for the event | Source 16 for the event |
| Unit | - | - | - | - | - | - | - | - | - | - | - | - | - | - | - | - |
| Exemple | https://www.rts.ch/info/suisse/6749453-le-chablais-et-le-bas-valais-restent-en-etat-d-alerte-face-aux-pluies.html | http://www.24heures.ch/vaud-regions/riviera-chablais/A-Monthey-les-secours-sont-prets-a-evacuer-les-riverains-de-la-Vieze/story/22490259 | http://www.24heures.ch/suisse/geneve-subit-grande-crue-arve-1935/story/10943703 | http://www.24heures.ch/vaud-regions/monthey-reveille-soulagee-evacuation-300-personnes/story/19307318 | http://www.lenouvelliste.ch/articles/valais/canton/inondation-a-st-gingolph-temoignages-du-president-et-de-restaurateurs-378561 | https://www.lemps.ch/Page/Uuid/b0e525de-f0c0-11e4-8a43-4ad205b10b56/Les_eaux_en_furie_dans_toute_la_Suisse | http://www.arcinfo.ch/articles/regions/neuchatel-et-littoral/inondations-a-cornaux-et-a-lignieres-378552 | http://www.romandie.com/news/Le-Chablais-fortement-touche-par-les-inondations/589780.rom | http://www.rts.ch/info/suisse/6749453-inondations-et-rivieres-en-crue-apres-les-fortes-pluies-.html | http://www.24heures.ch/suisse/suisse-romande/certaines-routes-valaisannes-fermees-cause-deluge/story/27182180 | https://www.rfj.ch/rfj/Actualite/Region/20150801-La-Roche-Saint-Jean-a-deux-doigts-de-l-inondation.html | http://www.20min.ch/ro/news/romandie/story/25748211 | - | - | - | - |
| Comment | - | - | - | - | - | - | - | - | - | - | - | - | - | - | - | - |
| Source | Google Alerts | Google Alerts | Google Alerts | Google Alerts | Google Alerts | Google Alerts | Google Alerts | Google Alerts | Google Alerts | Google Alerts | Google Alerts | Google Alerts | Google Alerts | Google Alerts | Google Alerts | Google Alerts |
| | 157 | 158 | 159 | 160 | 161 | 162 | 163 | 164 | 165 | 166 | 167 | 168 | 169 | 170 | 171 | 172 |



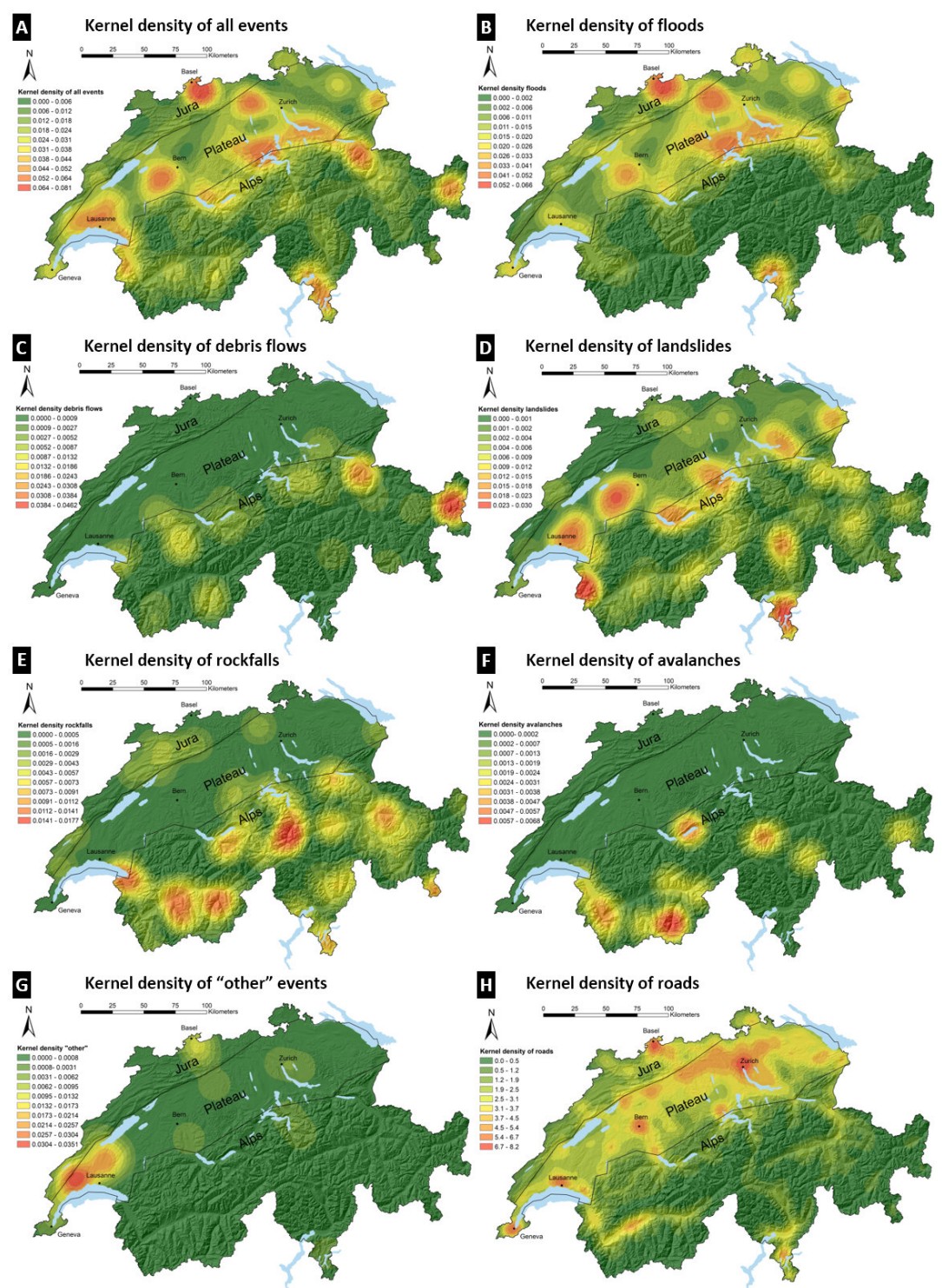


Figure 2-SM: Kernel density maps. Search radius for events: 20 km. Search radius for road network: 10 km. The
results were classified using 10 classes with the Jenks natural breaks method. A: All events; B: Floods; C:
Debris flows; D: Landslides; E: Rockfalls; F: Avalanches; G: "Other"; H: Roads. Hillshade and map ground
sources: Swisstopo.


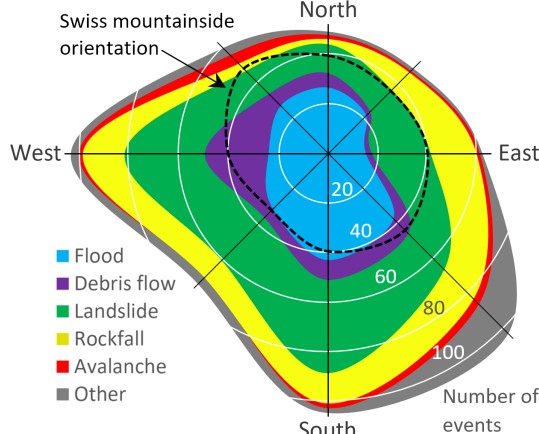


*Figure 3-SM: Slope orientation distribution of natural hazard events on the Swiss transportation network from*
*2012 to 2016. The relative distribution of Swiss mountainside orientation is shown by the black dashed line.*

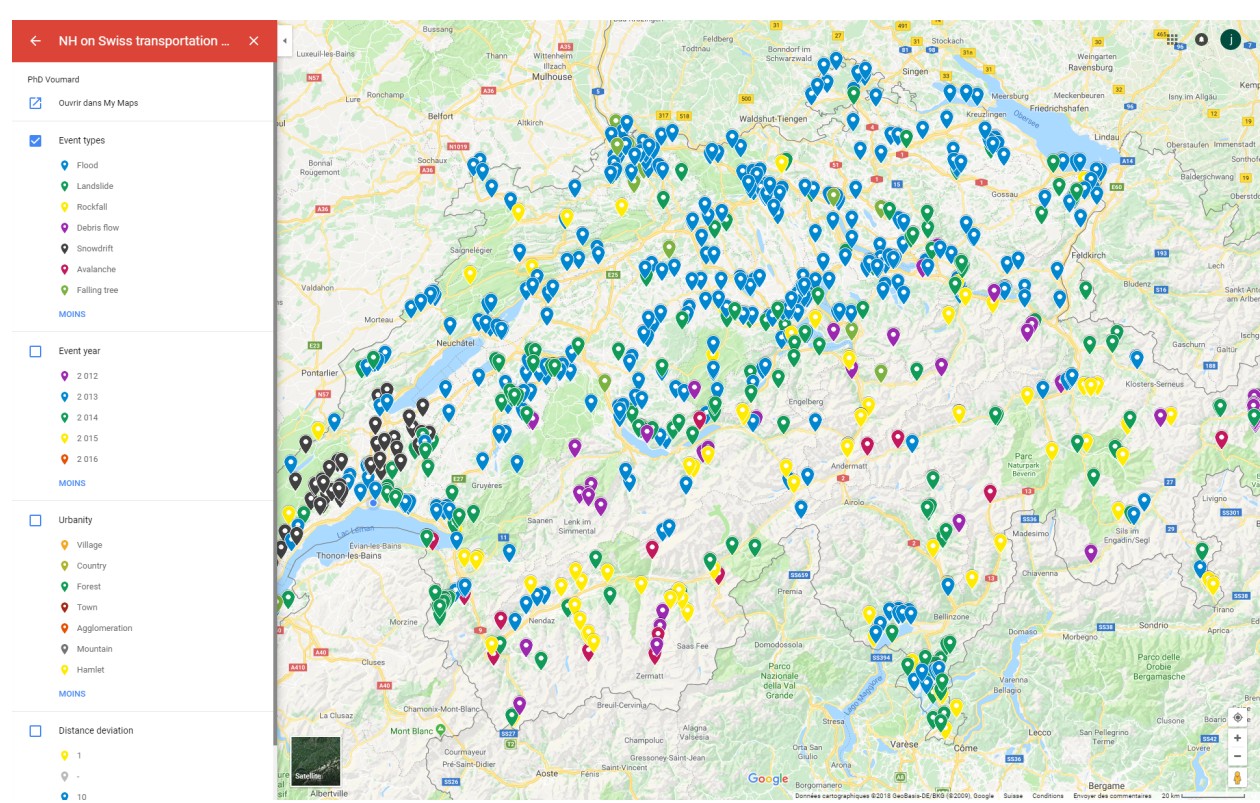


*Figure 4-SM: Database on Google Maps. Available at (last accessed: 25 January 2018):*
*https://www.google.ch/maps/@46.7199391,7.1246016,8z/data=!4m2!6m1!1s1qtu6LEYum-*
*7ghpPg9WWzWwgPHYA?hl=fr, last access: 25 January 2018.*

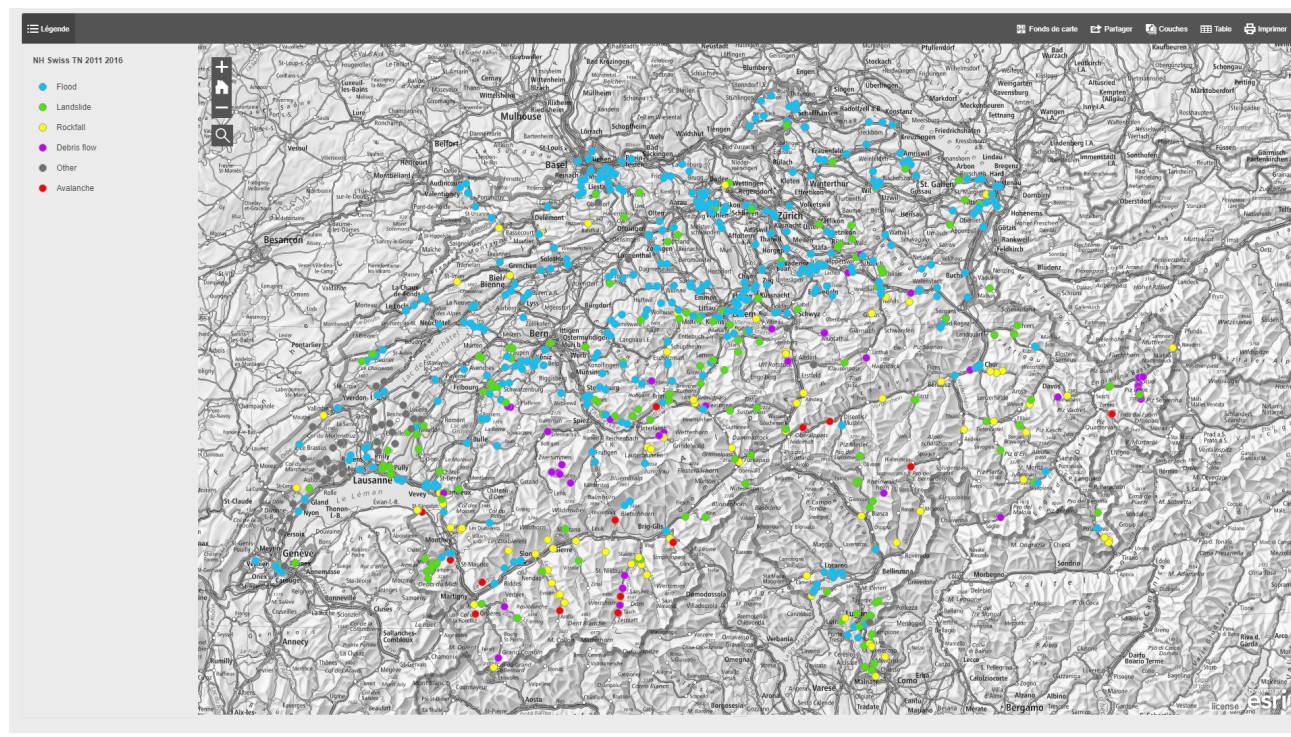


*Figure 5-SM: Database on ArcGIS online. Available at (last accessed: 25 January 2018):*
*http://unil.maps.arcgis.com/apps/MapTools/index.html?webmap=34ee3eb719a647889abd34175969d781, last*
*access: 25 January 2018.*