# Peer review of "1 Supplementary material"

_Natural Hazards and Earth System Sciences, 2018_

## Referee Comment (RC1) · H. J. Laimer (Referee) · 13 Feb 2018

The topic of the paper is definitely important for researchers in the field of applied earth science and construction or traffic engineers: small events are underrepresented in natural hazard research for several reasons but cause ever greater economic losses. The authors are trying to make the scientific community aware of the need to deal with the problem.

Therefore they collected online reports on natural hazard events affecting transportation networks in Switzerland. This approach could certainly be criticised for different reasons as data integrity or completeness, but the authors of course are aware of these problems. I think it is nearly the only way to get fast access to nationwide event data,

particularly since infrastructure operators often have reservations against publishing their data.

The declared objective of the authors is to help decision makers to minimise the impact of natural hazards (l. 83 – 85). I therefore recommend offering some suggestions for ways in which infrastructure operators could be assisted in order to better illustrate the benefit of the new database.

The factors of influence mentioned in the results chapter are not new, however, the paper provides essential statistical proofs!

I see only chapter 4.3.3 - Time of day and hourly distribution rather critically, because the time of event notification very frequently does not match with the real event time.

The reason for the high proportion of landslides on rail tracks can not only be found in bad embankment construction (l. 342 – l. 343). Railways have higher exposure to landslides than other line structures because of their grade limitations. Rail tracks require a balanced gradient ratio, therefore they must be run along the valley sides over far distances. This requires long and steep cutslopes.

There is a separate chapter 4.5.5 - Deviation length for roads. What about alternative routes for trains? Are there any informations on this issue? I suppose it is very difficult to get appropriate data.

I can hardly believe that highways are proportionally more vulnerable than other roads (l. 364 – 365). Is it not rather the case that small events on minor roads (e. g. non-public forest roads) are underrepresented in the database? The discussion chapter 5. 2. 1 contains a detailed outline of this problem (in particular l. 580 – 581).

The authors dare to the extremely sensitive subject of damage costs. It is difficult to get reliable data for direct costs, for indirect costs this is an almost impossible task. Costs per square meter (small event 100 m$^2$, middle event 200 m$^2$, large event 300 m$^2$) might seem unusual to infrastructure operators, but it could be a good approach to gain a

nationwide overview.

The figures are readable and helpful, a clear graphic visualization of the results.

The relevant articles and sources were quoted conscientiously.

-syntax and grammarconsistent thousands separators (e.g. 5.000) l. 24 . . .the database is imperfect because of. . . l. 48 . . .than for. . . l. 55, l. 58, l. 974 . . .Tchögl 2006, Tschögl et al. 2006 l. 269 . . .bad weather events l. 297 . . .and by the. . . l. 297 . . . precipitation. . .falls as snow l. 304 . . .to occur. . . l. 316 . . . 6 pm? l. 343 . . .earthy. . . unsuitable fill material l. 425. . . missing punctuation l. 440 – l. 442 and l. 534 – l.537 show a repeated text passage. l. 456. . .event mass? l. 464. . .before impacting. . . l. 539. . .debris flows l. 603. . .over the years l. 612. . .represents a certain l. 618. . .an impact. . . l. 631. . .word repetitions l. 669. . .without sufficient knowledge of natural hazards l. 693. . .have such an event database l. 695. . .Even if. . . l. 702. . .depends on. . . l. 744. . .railway tracks

---

## Referee Comment (RC2) · H. J. Laimer (Referee) · 21 Feb 2018

From the viewpoint of the geotechnical subdivision of an infrastructure operator I fully agree with the paper's main suggestion: focus on maintenance, since exhaustive protective measures are financially and technically not feasible. This includes not only constructional maintenance measures, but also (for instance) geomorphological mapping in the assessment of earthworks or geological investigations in rock slopes to optimize rock scaling intervals.

---

## Short Comment (SC1) · 21 Feb 2018

**Response to H. J. Laimer comments ( 1ˢᵗ Referee)**

*The authors would like to thank the first referee for his attentive lecture of the manuscript and his valuable comments and constructive suggestions.*

*We are waiting for the second referee comments before giving the revised text.*

**Referee comments**

The topic of the paper is definitely important for researchers in the field of applied earth science and construction or traffic engineers: small events are underrepresented in natural hazard research for several reasons but cause ever greater economic losses. The authors are trying to make the scientific community aware of the need to deal with the problem.

Therefore they collected online reports on natural hazard events affecting transportation networks in Switzerland. This approach could certainly be criticised for different reasons as data integrity or completeness, but the authors of course are aware of these problems. I think it is nearly the only way to get fast access to nationwide event data, particularly since infrastructure operators often have reservations against publishing their data.

The declared objective of the authors is to help decision makers to minimise the impact of natural hazards (l. 83 – 85). I therefore recommend offering some suggestions for ways in which infrastructure operators could be assisted in order to better illustrate the benefit of the new database.

*Answer: Our main suggestion (added in the paragraph at the end of the conclusion):*

*Since the risk reduction on a track is difficult because of it is too expensive to add protective measures on the all track section as small events can occurred almost everywhere, we must reflect in terms of traffic accessibility at a local scale. Robustness of the network must be increased by maintaining or creating alternatives tracks in order to avoid as possible traffic restriction and indirect damages. Direct damage resulting from small events are not huge but their indirect damages can be important and expensive for a region. To maintain, to improve or to create emergency accesses like forest tracks in case of road closure can avoid isolating villages.*

The factors of influence mentioned in the results chapter are not new, however, the paper provides essential statistical proofs!

I see only chapter 4.3.3 - Time of day and hourly distribution rather critically, because the time of event notification very frequently does not match with the real event time.

*Answer: We agree that the time of event from the medias articles must not be considered as the strict truth. However, we believe that they correspond more and less to the reality, particularly for time of event occurring during the day because time of event from police services are generally trusty, train drivers must generally note on their book the time of a track restriction, event occurring on road with a certain among of traffic has different witnesses as well as event occurring on street in a urban area. Time of events occurring during the night or on track with little traffic (forest road) effectively match rarely with the real event time. We added a sentence to mention this fact at the end of paragraph 4.3.3..*

The reason for the high proportion of landslides on rail tracks can not only be found in bad embankment construction (l. 342 – l. 343). Railways have higher exposure to landslides than other line structures because of their grade limitations. Rail tracks require a balanced gradient ratio, therefore they must be run along the valley sides over far distances. This requires long and steep cutslopes.

*Answer: Thank you for this important and correct input. We have added text about this at the end of paragraph 4.4.1..*

There is a separate chapter 4.5.5 - Deviation length for roads. What about alternative routes for trains? Are there any informations on this issue? I suppose it is very difficult to get appropriate data.

*Answer: We have estimated the deviation length for railways but as they are less pertinent as deviation length on roads and that we had to reduce the manuscript length, we do not keep the dedicated paragraph in the last version of the manuscript. We see three different ways to estimate deviation length of train closures: 1) To compute the deviation distance on train track between the two stations on both sides of the closure (= no replacement buses service); 2) To evaluate the deviation distance on road track between the two stations on both sides of the closure (= replacement buses); 3) To compute the real deviation distance during a event (on road with bus or on railway if no replacement service). We have evaluated the distance on train track (solution 1 above). The average distance of the 27 computed deviations was estimated at 65 km. For 72 events on railways, it was not possible to have a deviation by train. In a general way, we believe that the increase of the travel duration in case of railway closures is more relevant for passengers than the distance of deviation itself. We have added some sentences about this problematic at the end of chapter 4.5.5.*

I can hardly believe that highways are proportionally more vulnerable than other roads (l. 364 – 365). Is it not rather the case that small events on minor roads (e. g. nonpublic forest roads) are underrepresented in the database? The discussion chapter 5. 2. 1 contains a detailed outline of this problem (in particular l. 580 – 581).

*Answer: Yes, events on non-public roads are underestimated in the database and we tried to discuss this in the chapter 5.2.1.*

The authors dare to the extremely sensitive subject of damage costs. It is difficult to get reliable data for direct costs, for indirect costs this is an almost impossible task. Costs per square meter (small event 100 m2 , middle event 200 m2 , large event 300 m2 ) might seem unusual to infrastructure operators, but it could be a good approach to gain a nationwide overview.

*Answer: Yes, direct costs are difficult to estimate (and indirect costs are effectively almost impossible to be estimated). Even different services (police, road service / railway company, communes, canton, Confederation, etc.) do not really know how much and for what they have paid to fix the track closure. Our proposition of costs per square meter is unperfect but it is simply and based on our experience of affected area superficies and based on the costs experience of a civil engineer. We tried to discuss those cost uncertainties and large cost variability according the event location in the chapter 5.3.*

The figures are readable and helpful, a clear graphic visualization of the results.

The relevant articles and sources were quoted conscientiously.

**-syntax and grammar-**

consistent thousands separators (e.g. 5.000) *Answer: Space for thousands separators added.*

l. 24 . . .the database is imperfect because of. *Answer: Rewritten with :" the database is imperfect because of the way it was built".*

l. 48 . . .than for. . . *Answer: Rewritten with: " than large".*

l. 55, l. 58, l. 974 . . .Tchögl 2006, Tschögl et al. 2006 *Answer: Replaced and rewritten with: "Tchögl et al. 2006".*

l. 269 . . .bad weather events *Answer: "Meteorological" has been deleted.*

l. 297-6 . . .and by the. . . *Answer: Replaced with "by the fact that...".*

l. 297 . . . precipitation. . .falls as snow *Answer: Rewritten with "precipitations in mountains fall as snow".*

l. 304 . . .to occur. . . *Answer: Rewritten with "They occur".*

l. 316 . . . 6 pm? *Answer:18 pm was replaced with 6 pm.*

l. 343 . . .earthy. . . unsuitable fill material *Answer: Rewritten with "soil embankments or unsuitable fill material".*

l. 425. . . missing punctuation *Answer: Dot added.*

l. 440 – l. 442 and l. 534 – l.537 show a repeated text passage. *Answer: Exactly! L. 533 to 539 were now deleted.*

l. 456. . .event mass? *Answer: Replaced with "the material that fell".*

l. 464. . .before impacting. . . _Answer: Rewritten with "before impacting a fallen_

l. 539. . .debris flows _Answer: Sentence now deleted._

l. 603. . .over the years _Answer: Rewritten with "over the years"._

l. 612. . .represents a certain _Answer: "S" added._

l. 618. . .an impact. . . _Answer: Rewritten with "an impact"._

l. 631. . .word repetitions _Answer: Rewritten with "that can not be compared."._

l. 669. . .without sufficient knowledge of natural hazards _Answer: Rewritten with "without sufficient knowledge"._

l. 693. . .have such an event database _Answer: Rewritten with "no interest to have such an event database"._

l. 695. . .Even if. . . _Answer: Rewritten with "Even if"._

l. 702. . .depends on. . . _Answer: Rewritten with "it depends on"_

l. 744. . .railway tracks _Answer: Rewritten with "railway tracks"._

---

## Referee Comment (RC3) · Anonymous Referee #2 · 23 Mar 2018

Summary: While it is clear that the database outlined in the manuscript represents a large amount of work and a potential contribution to the discussion of natural hazards impacting transportation networks, there is significant reworking that is required before this manuscript could be considered acceptable for publication. The authors present a new database developed for hazards impacting transportation networks within Switzerland, however, the paper requires major reorganization, clarifications in the methodology and interpretation of results, substantial shortening, and significant improvement in the language and grammar before it can be published. Below are some major comments and remaining questions about the manuscript that may help to clarify and improve the overall article:

Database compilation methodology:

[Figure]

It is not clear what the authors deems as the minimum impact or threshold of when a natural hazard event is considered. The authors cite that a fraction of their database has no affect on the roadway, trail or railway in terms of disruption, etc. So why is it included in the database? The authors first must be clear what they are setting as the minimum threshold for being included in the database. It is also not clear from the current presentation of the methodology how the volume of the events or timing of events are determined. There are also variations in the reporting of the date range the database is compiled from, citing 2011-2016 and 2012-2016 throughout the manuscript. The methodology of using Google Alerts is reasonable, however, since this practice was only started in 2014, there are clear discrepancies between the number of events obtained before this practice was adopted and after. The authors specifically cite the change in number of hazards reported (a 2 fold increase from before and after 2014!). The comparison to the Canton of Vaud dataset for hazards is confusing. It appears that it could be a useful dataset from which to compare but the authors are not clear about what specifically the differences are, they merely report the number of events. It may be helpful to look more into the differences in types of events between these two databases to provide more quantitative metrics on potential biases with the database presented in this paper. The determination of event cost is interesting and could be a valuable contribution to the paper if it was further substantiated in its own section. In the current way it is presented it is a bit unclear how robust or realistic the assigned cost values are.

Presentation of database statistics:

While it is important to highlight the different characteristics of the database, I feel there is no need to present every aspect of this database as percentages based on the type of hazard and attribute being considered. This makes the paper much longer than it needs to be and in my opinion does not add value. I would recommend the authors significantly reduce the number of figures into several key attributes that the authors feel best describe the unique aspects or findings of this database and include any

other metrics or distributions the authors feel are relevant in supplementary material (or not include them at all). The manuscript presents many statistics without much or any discussion of why it is significant, what is suggests about the nature of the hazard for the specific attribute collected or cites other sources of information or analysis that supports the findings. Many of the sections outlining the statistics end with one to two sentences that should be significantly strengthened to clearly summarize the points being made.

Overall structure and content:

The paper overall is currently much longer than it needs to be with far too many figures. I recommend the paper be drastically shortened to highlight the most salient points in the database and take more time in the discussion section to outline how this database can be used or should be interpreted to make clear points about the role of small natural hazards on transportation networks in Switzerland. The discussion section at present seems to go off on several apparent tangents about the limitations and lack of interest in this type of database; however, the authors could instead use this opportunity to outline the value of the database despite its limitations and the potential applicability of the database within communities of interest. Finally, there are a large number of grammatical and language errors in the manuscript that must be corrected.

---

## Editor Comment (EC1) · F E Taylor (Editor) · 12 Apr 2018

Dear Jerome and Co-Authors,

Thank you to reviewers Hans-Joerg Laimer and Annonymous Reviewer 2 for their helpful input and feedback. Both reviewers commented on the value of databases like these and encourage you to bring out the utility of your work in the discussion section.

Reviewer 2 has made some important comments about the methodology that must be addressed, possibly requiring some reanalysis of the database or a clearer justification for the methods chosen and findings made from an uncertain database.

There are some issues throughout with the writing such as English language, consistency and lack of references to support some statements made.

Reviewer 2 has also commented that the paper could be shortened by reducing or combining the number of figures, and focusing more on the key results rather than describing all results. I believe this will make a stronger, more focused paper.

With these medium to major level revisions, the paper will make a good contribution to NHESS, as highlighted by reviewer 1.

I look forward to reading the revised manuscript.

Kind regards,

Faith

---

## Author Comment (AC1) · 23 May 2018

**Response to the Anonymous Referee #2**

*The authors would like to thank the second referee for his attentive lecture of the manuscript and his valuable comments and constructive suggestions. Answers to the questions are in italic.*

**Referee comments**

Database compilation methodology

- The authors cite that a fraction of their database has no affect on the roadway, trail or railway in terms of disruption, etc. So why is it included in the database?

  *All collected events have generated traffic disruptions. If 18% of events have generated no damage on tracks, they have, at least, generated traffic disruptions (please see next question/answer).*

- It is not clear what the authors deems as the minimum impact or threshold of when a natural hazard event is considered. The authors first must be clear what they are setting as the minimum threshold for being included in the database.

  *A **sentence** in Section 3, data and methods, **was added**: "The minimum threshold for being included in the database is a traffic disruption (for example, a large velocity reduction) for at least 10 minutes following a natural hazard event that have reached to a transportation track."*

- It is also not clear from the current presentation of the methodology how the volume of the events or timing of events are determined.

  ***Sentences** in Section 3, data and methods, **were added**: "Data about date, location, event characterization and damage come from the online press articles". "Images from the press articles are used to estimate many attributes as the event classification, the track damage and the volume of the deposit material if it is not given in the press article."*

- There are also variations in the reporting of the date range the database is compiled from, citing 2011-2016 and 2012-2016 throughout the manuscript.

  *Dates were effectively different along the manuscript. They were corrected: 2012-2016.*

- The methodology of using Google Alerts is reasonable, however, since this practice was only started in 2014, there are clear discrepancies between the number of events obtained before this practice was adopted and after. The authors specifically cite the change in number of hazards reported (a 2 fold increase from before and after 2014!).

  *Exact, the methodology has its limits. Expect for floods (and landslides), the difference of number of events does not increase so much since the use of Google Alerts. **Sentences were added**: " Google Alerts permits mainly to improve the event collection of floods. Moreover, the total number of event increases year after year, even after the use of Google Alerts (mid 2014) because of the increase of floods disruptions (Figure 25 of submitted manuscript; Figure 6F of the new corrected version). This shows that the use of Google Alerts is not fully responsible of*

*the yearly increase of number of events. Those numbers depend strongly to the weather conditions that are different each year".*

- The comparison to the Canton of Vaud dataset for hazards is confusing. It appears that it could be a useful dataset from which to compare but the authors are not clear about what specifically the differences are, they merely report the number of events. It may be helpful to look more into the differences in types of events between these two databases to provide more quantitative metrics on potential biases with the database presented in this paper.

  *In order **to shorten the document**, this section "Comparison with the Canton of Vaud database" was removed.*

- The determination of event cost is interesting and could be a valuable contribution to the paper if it was further substantiated in its own section. In the current way it is presented it is a bit unclear how robust or realistic the assigned cost values are.

  *As the direct damage cost assessment is difficult and can range according the event features, the direct cost determination is, above all, a cost comparison tool of the different damage classes. **Sentences were added** at the end of section 3, Data and methods: "Since direct damage costs are difficult to assess (this is event more true for indirect damage costs), the proposed methodology to determine them must be considered, above all, as a tool to compare the costs of the different damage classes. The cost values should not be considered as true costs for all events but as a order of magnitude of the projected costs (please see also section 5.4)". Furthermore, the section 5.4, direct damage cost estimation, should discuss about this challenge of cost determination. Even if they are not easy to assess them, direct damage costs may be important giving a order of magnitude of direct costs induced by all events that affect the transportation networks.*

Presentation of database statistics

- While it is important to highlight the different characteristics of the database, I feel there is no need to present every aspect of this database as percentages based on the type of hazard and attribute being considered. This makes the paper much longer than it needs to be and in my opinion does not add value.

  *Concerning the percentage given in the results, the opinions of the authors is that both number of events and percentages are a source of information that must not be removed. We realize that percentage and number of events do not make the text simpler but we consider that it can be important for the reader to know the absolute values of the given data. We let the Editor select to keep or not the % and absolute values.*

- I would recommend the authors significantly reduce the number of figures into several key attributes that the authors feel best describe the unique aspects or findings of this database and include any other metrics or distributions the authors feel are relevant in supplementary material (or not include them at all).

  *We have **deleted figures** about sinuosity and affected population. All remaining **figures were regroup** in few images.*

- The manuscript presents many statistics without much or any discussion of why it is significant, what is suggests about the nature of the hazard for the specific attribute collected or cites other sources of information or analysis that supports the findings. Many of the sections outlining the statistics end with one to two sentences that should be significantly strengthened to clearly summarize the points being made.

  *The main issue of this paper is to present the collected data and to discuss those of them that have global interest (and not a interest only for Swiss people). It is a synthesis of the database and not a detailed discussion about collected results that could be part of a other paper. We have chose not to discuss in detail all presented data because it would be much longer and because there is too much material for one paper. With the presented data, the reader have an detailed overview of the features of NH that have affected the Swiss transportation networks last 5 years. He can compare freely those results with other sources if he so wishes, especially with the detailed results that can be found in the Tables of the Supp. Material.*

Overall structure and content

- The paper overall is currently much longer than it needs to be with far too many figures. I recommend the paper be drastically shortened to highlight the most salient points in the database…

  *Right! The paper as too long. He **was reduced of 28%** although it contains now **new analysis** as the **risk ratios** and the **kernel density maps**. All initial figures were regrouped in three images. Some sections were deleted as the "track sinuosity" or the "population affected".*

- … and take more time in the discussion section to outline how this database can be used or should be interpreted to make clear points about the role of small natural hazards on transportation networks in Switzerland. The discussion section at present seems to go off on several apparent tangents about the limitations and lack of interest in this type of database; however, the authors could instead use this opportunity to outline the value of the database despite its limitations and the potential applicability of the database within communities of interest. Finally, there are a large number of grammatical and language errors in the manuscript that must be corrected.

  *We have **added a short section** about the value of the database in the risk management in Switzerland: "Risk management in Switzerland may therefore be improved with such a database. For examples, it shows the important alternative ways to bypass the obstacles. We have highlighted that for one quarter of events, there were no deviation routes. This proportion is high and must be reconsidered by the authorities. It is evident that to protect all swiss tracks against natural hazard processes would me much to expensive. Thus, it is essential to guaranty alternative tracks and to fund protective measures with the best ratio cost / risk reduction. Minor roads often belong to the municipalities which does not have a great interest to maintain them. The Cantons and the Confederation would be advised to participate or even to take over the maintenance of some of them that can be vital in case of closure of main roads or railway tracks. This is particularly appropriate in transportation corridor when the minor road is located on the other valley side than the major road. With its national scale, this database helps to consider the risk of transportation networks tracks more from a network perspective than from a track scale."*

*Section about the limitation of the database were reduced but kept, as it is important to know its limitations.*

---

## Author Comment (AC2) · 23 May 2018

The authors totally agree with the RC2 comment. The maintenance of the infrastructure is a key element in the risk management.
* * *

---

## Author Response (AR1)

[revised manuscript text omitted]

75969d781, last access: 25 January 2018

---

## Referee Report (RR1)

The initial criticisms have been integrated into the current version of the paper. The shortened version is easier to read and thus easier to understand.

The benefit of the new database is adequately explained in chapter 6 – Conclusion and perspectives.

The reasons for the high proportion of landslides on rail tracks were completed.

Chapters 5 and 6 were completely revised.

The introductory sentences of. 5.2 are very important even if they are only marginally involved with the topic. That would be worth to be discussed separately (in another paper).

I don't see any need to change the contents/figures any more.

A second reworking with regard to English grammar and careless mistakes would improve the quality of the

paper.

For example (incomplete list):

Line 188: debris flows have occured?

Line 331: Swiss cities?

Line 432: depends on

Line 436: on Gotthard highway

Line 502: in Austria

Line 505: rockfall events (or rockfalls) do…

Line 636: would be

Line 647: Our analysis is?

---

## Author Response (AR2)

**Minor revisions, author response**

- All values of results are now given firsty in percent and secondly in number.
- English language has been professionally corrected by AJE.